# Human phospho-signaling networks of SARS-CoV-2 infection are rewired by population genetic variants

Diogo Pellegrina[1,†] iD, Alexander T Bahcheli[1,2,†] iD, Michal Krassowski[3] iD & Jüri Reimand[1,2,4,*] iD

## Abstract

SARS-CoV-2 infection hijacks signaling pathways and induces protein–protein interactions between human and viral proteins. Human genetic variation may impact SARS-CoV-2 infection and COVID-19 pathology; however, the genetic variation in these signaling networks remains uncharacterized. Here, we studied human missense single nucleotide variants (SNVs) altering phosphorylation sites modulated by SARS-CoV-2 infection, using machine learning to identify amino acid substitutions altering kinase-bound sequence motifs. We found 2,033 infrequent phosphorylation-associated SNVs (pSNVs) that are enriched in sequence motif alterations, potentially reflecting the evolution of signaling networks regulating host defenses. Proteins with pSNVs are involved in viral life cycle and host responses, including RNA splicing, interferon response (TRIM28), and glucose homeostasis (TBC1D4) with potential associations with COVID-19 comorbidities. pSNVs disrupt CDK and MAPK substrate motifs and replace these with motifs of Tank Binding Kinase 1 (TBK1) involved in innate immune responses, indicating consistent rewiring of signaling networks. Several pSNVs associate with severe COVID-19 and hospitalization (STARD13, ARFGEF2). Our analysis highlights potential genetic factors contributing to inter-individual variation of SARS-CoV-2 infection and COVID-19 and suggests leads for mechanistic and translational studies.

**Subject Categories** Computational Biology; Microbiology, Virology & Host Pathogen Interaction; Post-translational Modifications & Proteolysis

**Mol Syst Biol. (2022) 18: e10823**

## Introduction

Severe acute respiratory syndrome coronavirus 2 (SARS-CoV-2) and the coronavirus disease 2019 (COVID-19) pandemic has caused millions of deaths worldwide and continues to evolve as more

pathogenic and infectious variants of the virus emerge. The clinical manifestations and outcomes of COVID-19 are complex, ranging from asymptomatic infection to fatal respiratory and multi-organ failure, as well as long-term symptoms after recovery. Risk factors of severe disease include advanced age, a weakened immune system, and pre-existing health conditions such as hypertension, diabetes, and obesity (Richardson *et al*, 2020). Ethnic and demographic patient characteristics that are at least partially explained by socio-economic factors also affect disease risk (Nicola *et al*, 2020; Pareek *et al*, 2020; Webb Hooper *et al*, 2020). Recent genome-wide association studies (GWAS) and exome-sequencing efforts of COVID-19 patients have shed light on human genetic variation contributing to SARS-CoV-2 disease burden and mortality rates, highlighting genes associated with ABO blood groups, antiviral pathways, and tyrosine kinase signaling (Covid-19 Host Genetics Initiative, 2021a; Severe Covid Gwas Group *et al*, 2020; Kosmicki *et al*, 2021a; Pairo-Castineira *et al*, 2021). However, GWAS findings often occur in intergenic regions where molecular mechanisms remain elusive, and rare variants potentially contributing to disease are challenging to detect. Thus, additional work is needed to find and interpret genetic variants contributing to SARS-CoV-2 infection and COVID-19 pathology.

Molecular interaction networks are perturbed by host–pathogen interactions and host disease mutations that either disable proteins in the networks or alter their interactions (Vidal *et al*, 2011). Phosphorylation is a key component of cellular signaling networks that acts as a reversible molecular switch controlling protein function and interactions. This post-translational modification (PTM) is conducted by kinases that recognize sequence motifs at protein phosphorylation sites (*i.e.*, phosphosites). SARS-CoV-2 infection alters phosphorylation networks in host cells by promoting casein kinase 2 (CK2) and mitogen-activated protein kinase (MAPK) pathways and inhibiting mitotic kinases, resulting in cell cycle arrest and cytoskeletal changes to favor virus pathology (Bouhaddou *et al*, 2020). Phosphorylation networks also control host anti-viral and immune responses to SARS-CoV-2 infection. For example, the TANK Binding Kinase 1 (TBK1) and IKK-epsilon kinase of the NF-κB

1 Computational Biology Program, Ontario Institute for Cancer Research, Toronto, ON, Canada
2 Department of Molecular Genetics, University of Toronto, Toronto, ON, Canada
3 Medical Sciences Division, Nuffield Department of Women's and Reproductive Health, University of Oxford, Oxford, UK
4 Department of Medical Biophysics, University of Toronto, Toronto, ON, Canada
*Corresponding author. Tel: +1 647-260-7983; E-mail: juri.reimand@utoronto.ca
†These authors contributed equally to this work

pathway initiate innate antiviral response by phosphorylating interferon regulatory factors (IRF) that regulate interferon genes (Sharma et al, 2003; Balka et al, 2020). Interferons activate JAK-STAT, p38 MAPK, and PI3K/Akt signaling pathways that are dysregulated in SARS-CoV-2 infection and COVID-19 (Appelberg et al, 2020; Bouhaddou et al, 2020; Matsuyama et al, 2020). SARS-CoV-2 proteins bind TBK1 to suppress host immune responses (Lei et al, 2020; Xia et al, 2020). NF-κB hyperactivation has been associated with cytokine storms in COVID-19 where excess production of proinflammatory cytokines can fatally damage the host (Hirano & Murakami, 2020). Besides host proteins, PTMs of virus proteins also contribute to the complexity of infection and host anti-viral responses (Mishra et al, 2021).

Genetic variants involved in human disease are known to alter kinase signaling networks. For instance, inherited disease mutations and somatic mutations in cancer driver genes frequently erase phosphosites or alter kinase binding motifs, potentially causing rewiring of kinase signaling (Li et al, 2010; Reimand & Bader, 2013; Creixell et al, 2015; Huang et al, 2018). Conversely, phosphosites have lower genetic variation in the human population, underlining the importance of conserved phosphosites in evolution (Li et al, 2010; Reimand et al, 2015). Therefore, human genetic variation of signaling networks responding to SARS-CoV-2 infection may contribute to a range of symptoms, disease severity, and long-term outcomes of COVID-19 patients.

We hypothesized that genetic variation in the phosphosites that are differentially phosphorylated in SARS-CoV-2 infection can alter kinase signaling interactions and thereby reveal mechanistic insights into the genes and pathways involved in infection and disease. We studied the gnomAD dataset (Karczewski et al, 2020a), the largest uniformly processed map of human genetic variation, and mapped missense single nucleotide variants (SNVs) to human protein phosphosites responding to SARS-CoV-2 infection (Bouhaddou et al, 2020). With a machine learning approach, we uncovered hundreds of phosphorylation-associated SNVs (pSNVs) that modify kinase-bound sequence motifs and potentially rewire kinase-substrate interactions. Our study helps decipher the role of human genome variation in virus responses and disease outcomes and enables advances in therapy and biomarker development.

## Results

### SARS-CoV-2-associated phosphosites are enriched in pSNVs that rewire kinase binding motifs

To examine the genetic variation in signaling networks responding to SARS-CoV-2 infection, we studied exome sequencing data of 124,748 individuals from 16 human populations in the gnomAD dataset (Karczewski et al, 2020a), focusing on 1,111,194 amino acid substitutions that were observed at least once per 10,000 individuals in at least one of the populations ($AF_{popmax} > 10^{-4}$). We defined phosphorylation-associated SNVs (pSNVs) as missense SNVs (i.e., amino acid substitutions) that occurred in 1,530 host phosphosites differentially phosphorylated in SARS-CoV-2 infection based on an earlier phosphoproteomics study (FDR < 0.05; 24 h timepoint) (Bouhaddou et al, 2020). We mapped pSNVs in flanking windows of ± 7 amino acid residues around phosphorylated residues

(Fig 1A). To predict the functional impact of pSNVs on protein phosphorylation, we evaluated the effects of pSNVs on 125 types of sequence motifs bound by kinases of 77 families. We used MIMP (Wagih et al, 2015), a machine learning method trained on kinase binding sequences that evaluates whether a pSNV disrupts an existing motif or creates a new motif relative to the reference protein sequence. Higher Bayesian posterior probabilities computed by MIMP reflect an increased likelihood of pSNVs to rewire sequence motifs.

We found 2,033 pSNVs in 987 SARS-CoV-2-associated phosphosites and 693 genes (Fig 1B, Dataset EV1). We assigned these to four classes based on their predicted functional impact. First, direct pSNVs (99 or 5%) replaced central phospho-residues and caused loss of phosphosites, likely representing the pSNV class of highest impact. Second, motif-rewiring pSNVs (410 or 20%) created or disrupted sequence motifs, potentially causing gains or losses of kinase binding at the phosphosites (MIMP posterior prob > 0.5) (Dataset EV2). The remaining two classes of pSNVs lacked functional predictions based on sequence analysis and included pSNVs annotated as proximal or distal to the closest phosphosite (1,527 or 75%). When combining direct and motif-rewiring pSNVs, potential disruption of phosphorylation through motif alterations or phospho-residue replacement was predicted for 354 pSNVs. 87 pSNVs induced new sequence motifs at SARS-CoV-2-associated phosphosites and were predicted to cause gain of phosphorylation by a specific kinase (Fig 1C). Interestingly, 60 pSNVs caused motif switching such that the pSNV simultaneously disrupted one motif and replaced it with another motif at the same phosphosite.

We evaluated the statistical significance of these functional pSNV annotations by randomly sampling human phosphosites as controls. Compared to all phosphosites in the human proteome, SARS-CoV-2-associated phosphosites were enriched in motif-rewiring pSNVs (410 pSNVs observed versus $321 \pm 21$ pSNVs expected; $P < 10^{-4}$) (Fig 1B). In contrast, direct pSNVs replacing central phospho-residues, as well as proximal and distal pSNVs lacking motif-based predictions, were found less frequently in SARS-CoV-2-associated phosphosites.

We examined the functional annotations of the phosphosites with pSNVs. Some pSNVs (213 or 10%) occurred at well-studied phosphosites bound by specific kinases, such as cyclin-dependent kinases, casein 2 kinases, checkpoint kinases, and MAP kinases (Fig 1D). In most cases, known binding sites of kinases were enriched in pSNVs compared to similar sets of phosphosites sampled from the proteome. Many of these kinases were also previously identified in the signaling network responding to SARS-CoV-2 infection (Bouhaddou et al, 2020), confirming our findings. Further, the SARS-CoV-2-associated phosphosites with pSNVs had significantly higher functional relevance scores (Ochoa et al, 2020) than other human phosphosites (median site score 0.44 versus 0.32 expected, Wilcoxon $P = 9.7 \times 10^{-111}$) (Fig 1E). These functional characteristics of mutated phosphosites add confidence to our predictions of pSNV impact.

We asked if the phosphosites and pSNVs associated with SARS-CoV-2 infection were also relevant to other virus infections. We studied differential phosphosites observed in infections of Herpes simplex virus type 1 (HSV-1) (Kulej et al, 2017) and human immunodeficiency virus 1 (HIV-1) (Greenwood et al, 2016) (FDR < 0.05) and compared these with SARS-CoV-2-associated phosphosites. The

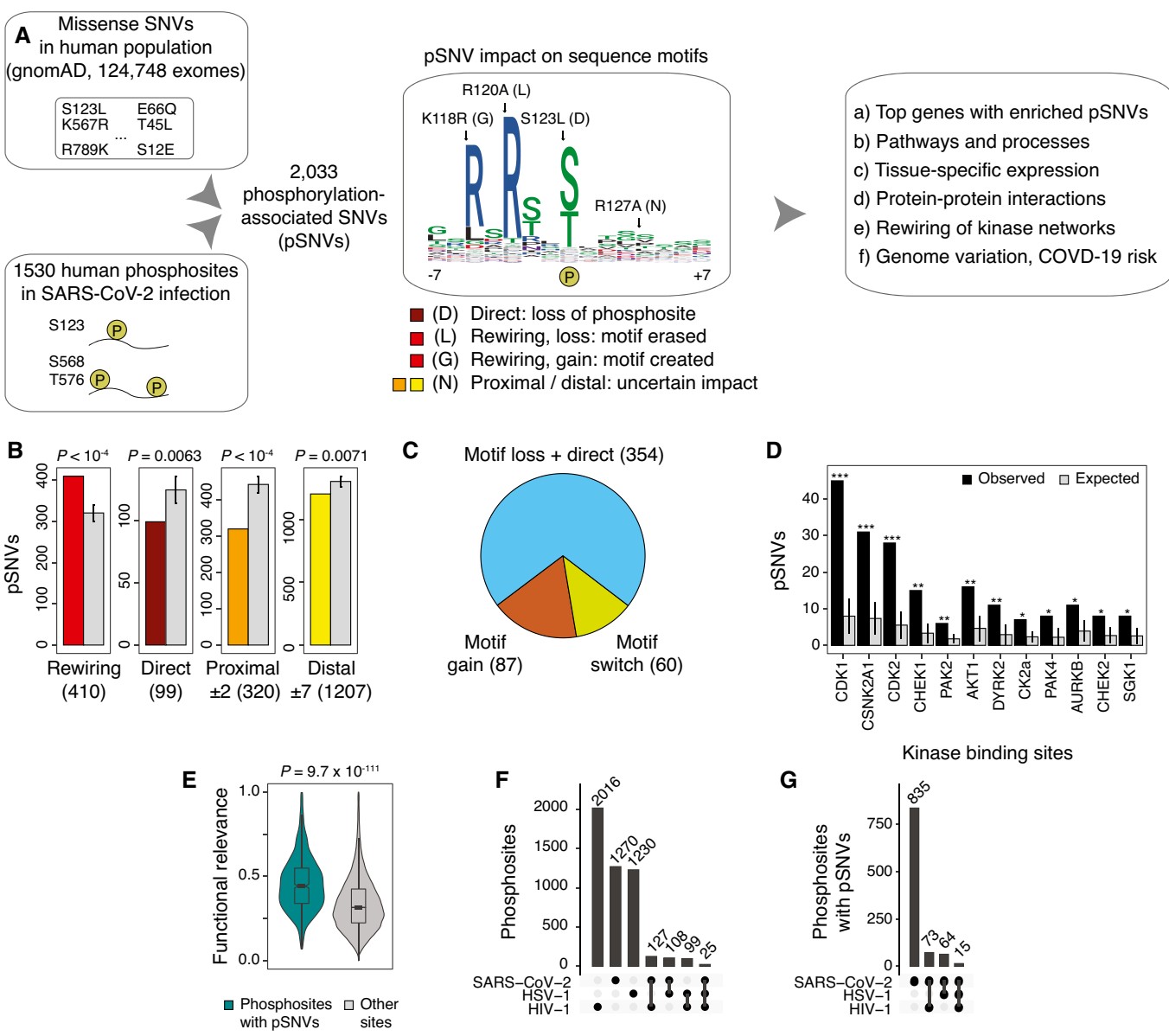

**Figure 1.  Phosphorylation-associated single nucleotide variants (pSNVs) in human phosphosites modified in SARS-CoV-2 infection.**

A   Overview of analysis. Missense SNVs in the human population (allele frequency $AF_{popmax} \geq 10^{-4}$ in gnomAD) were mapped to host protein phosphosites significantly differentially phosphorylated in SARS-CoV-2 infection (Bouhaddou et al, 2020). Sequence motif analysis was used to evaluate pSNV impact and to predict gains and losses of kinase binding sites. Genes, pathways, and molecular interaction networks with frequent pSNVs were analyzed.

B   pSNVs of four functional classes were studied: (1) motif-rewiring pSNVs altering sequence motifs in kinase binding sites (MIMP posterior *prob* > 0.5), (2) direct pSNVs altering phosphorylated residues, and (3–4) proximal and distal pSNVs with uncertain impact ($\pm$ 1–2 and $\pm$ 3–7 residues, respectively). Expected numbers of pSNVs (grey) represent control phosphosites sampled randomly from the proteome. pSNV counts and *P*-values of permutation tests are shown (10,000 permutations). Error bars show $\pm$ 1 SD.

C   Pie chart shows pSNVs predicted to cause loss of phosphosites (*i.e.*, pSNVs substituting phospho-residues and/or disrupting sequence motifs; blue), gain of phosphosites (*i.e.*, pSNVs inducing new sequence motifs; orange), and motif switches (pSNVs erasing one motif and inducing another motif).

D   Enrichment analysis of pSNVs at high-confidence binding sites of kinases. Bar plot shows the kinases whose known binding sites have significantly enriched pSNVs. The expected values were derived similarly to panel B (permutation test with 10,000 permutations). Error bars show $\pm$ 1 SD (*$P$ < 0.05; **$P$ < 0.01; ***$P$ < 0.001).

E   Violin plot of functional relevance scores (Ochoa et al, 2020) of SARS-CoV-2-associated phosphosites ($n$ = 895) and other human phosphosites as controls ($n$ = 70,833). Wilcoxon rank-sum $P$-value is shown.

F, G   Upset plots show the extent of overlap between phosphosites modified by SARS-CoV-2 and two other viruses (HSV-1 (Kulej et al, 2017), HIV (Greenwood et al, 2016)), and their overlaps with the SARS-CoV-2-associated phosphosites with pSNVs.

three sets of phosphosites were relatively distinct as 260 (17%) of the 1,530 SARS-CoV-2-associated phosphosites were shared with other infections. There were 152 SARS-CoV-2-associated sites with pSNVs that were shared with other infections (10% of 1,530), which were fewer than expected from chance alone (168 expected, Fisher's exact $P = 0.027$) (Fig 1F and G). Thus, we expect that our findings of pSNVs are mostly specific to SARS-CoV-2 infection.

Enrichment of motif-rewiring, functional pSNVs suggests that the signaling network responding to SARS-CoV-2 infection may vary in the human population such that certain kinase-substrate interactions are gained or lost in some individuals. This may reflect positive evolutionary selection in adaptation to viral infection and potentially cause variation in SARS-CoV-2 infection and COVID-19 disease course.

### Top genes with pSNVs are involved in RNA splicing, virus infection, and host immune response

We studied the genes with the most impactful pSNVs in SARS-CoV-2-associated phosphosites by evaluating the null hypothesis that none of the pSNVs per gene caused sequence motif alterations in the resulting protein. We computed significance scores for all genes by assigning more weight to the genes where multiple pSNVs altered kinase binding motifs or replaced central phospho-residues.

Gene prioritization revealed 77 genes with pSNVs predicted to alter protein phosphorylation through central residue or sequence motif alterations ($FDR < 0.1$), collectively including nearly one third of all pSNVs (Fig 2A, Dataset EV3). RNA splicing and anti-viral defense responses were prominent among the functions of top genes. The two most significant genes, *SRRM2* and *BUD13*, encode subunits of the spliceosome complex and include 83 and 11 pSNVs, respectively ($FDR < 10^{-23}$). SRRM2 is involved in human immunodeficiency virus (HIV) pathogenesis via alternative splicing (Wojcechowskyj *et al*, 2013). BUD13 regulates the antiviral transcription factor *IRF7* and interferon I response upon RNA-virus infection (Zhang *et al*, 2018; Frankiw *et al*, 2019). Interferon signaling triggers the first response of host cells to viral infection and activates the immune system of adjacent cells to suppress viral replication. The host RNA splicing machinery is targeted by SARS-CoV-2 proteins to disrupt splicing and impair host gene translation (Banerjee *et al*, 2020; Finkel *et al*, 2021). Another related candidate gene *BCLAF1* encodes a pro-apoptotic transcription and splicing factor and includes 18 pSNVs ($FDR = 4.2 \times 10^{-2}$). Apoptosis is triggered in infected host cells as a final form of cellular defense, while activation of anti-apoptotic pathways is a strategy to maximize viral replication (Ostaszewski *et al*, 2021). BCLAF1 is also involved in type I interferon signaling and regulates antiviral gene expression upon virus infection (Qin *et al*, 2019). The essential role of BCLAF1 in lung development (McPherson *et al*, 2009) and expression in lung cells suggests its activity in airway tract tissues affected by SARS-CoV-2 infection. Thus, some top genes with pSNVs are involved in core host cellular processes of virus infection and host immune response.

### pSNVs in TRIM28 associate with immune response regulation

We highlighted *TRIM28* with five pSNVs as a gene of interest ($FDR = 0.042$) (Fig 2B). *TRIM28* encodes transcriptional repressor that increases interferon beta and pro-inflammatory cytokine production in response to avian virus infection in lung epithelial cells through phosphorylation of S473 (Krischuns *et al*, 2018). This phosphosite is also modified upon SARS-CoV-2 infection in Vero6 cells (Bouhaddou *et al*, 2020).

One direct pSNV in TRIM28 affects the S473 phosphosite: the amino acid substitution S473L removes the central phospho-residue and causes loss of phosphorylation. The phosphosite S473 is bound by checkpoint kinases (CHEK1/2) and the MAPKAP2 kinase in DNA damage response and interferon activation (Hu *et al*, 2012; Krischuns *et al*, 2018). In agreement with these known kinase binding sites, the sequence motifs of CHEK1, CHEK2, and MAPKAP2 are disrupted by the pSNV (MIMP posterior $prob \geq 0.99$) (Fig 2C). Another pSNV in TRIM28, R472C, occurs upstream of the S473 phosphosite; however, unlike S473L, this substitution does not cause significant motif alterations. A recent study showed that knockdown of *TRIM28* activates ACE2 and leads to increased host cell entry of SARS-CoV-2 (Wang *et al*, 2021). TRIM28 expression also correlates with interferon levels, being lower in children with severe COVID-19 disease and multisystem inflammatory syndrome (MIS-C) compared to uninfected children and those with mild disease (Tovo *et al*, 2021). The pSNVs in TRIM28 may alter the signaling of this protein and suppress immune response. Further study of TRIM28 and these pSNVs may offer mechanistic insights to SARS-CoV-2 infection and COVID-19.

### Genes with frequent pSNVs are broadly expressed and enriched in Ras/Rho signaling, RNA splicing, and ER transport

To interpret the human genetic variation of signaling networks responding to SARS-CoV-2 infection, we asked if the genes with frequent pSNVs converged to biological processes, molecular

**Figure 2. Top genes, pathways, and tissue-based expression patterns with SARS-CoV-2-associated pSNVs.**

A  Top genes with pSNVs predicted to alter phosphosites modified by SARS-CoV-2 infection ($FDR < 0.1$). Genes are scored by the probability of at least one pSNV altering SARS-CoV-2-associated phosphosites.

B  TRIM28, a transcriptional and epigenetic regulator involved in innate immune response, includes five pSNVs in two phosphosites: S473 phosphorylated by CHEK and MAPKAP2 kinases, and S50 with no known kinase.

C  The pSNV S473L in TRIM28 substitutes the phospho-residue and disrupts a CHEK1 sequence motif with a high-confidence prediction of motif alteration (MIMP posterior $prob > 0.99$).

D  pSNVs are enriched in biological processes and molecular pathways (ActivePathways, $FDR < 0.2$). The enrichment map shows pathways and processes as nodes that are connected via edges if the pathways share many genes. Subnetworks are annotated with representative pathways and processes, and related genes with pSNVs are listed (gene $FDR < 0.25$).

E  Genes with frequent pSNVs are often expressed in various human tissues, as detected by enrichment analysis of expression signatures of the Human Protein Atlas (ActivePathways, $FDR < 0.2$). Bar plot (left) shows all the genes with pSNVs in each tissue. Grid plot (right) shows the top genes with pSNVs (panel A) and the tissues with corresponding gene expression. Color gradient corresponds to the $FDR$ values in panel A.

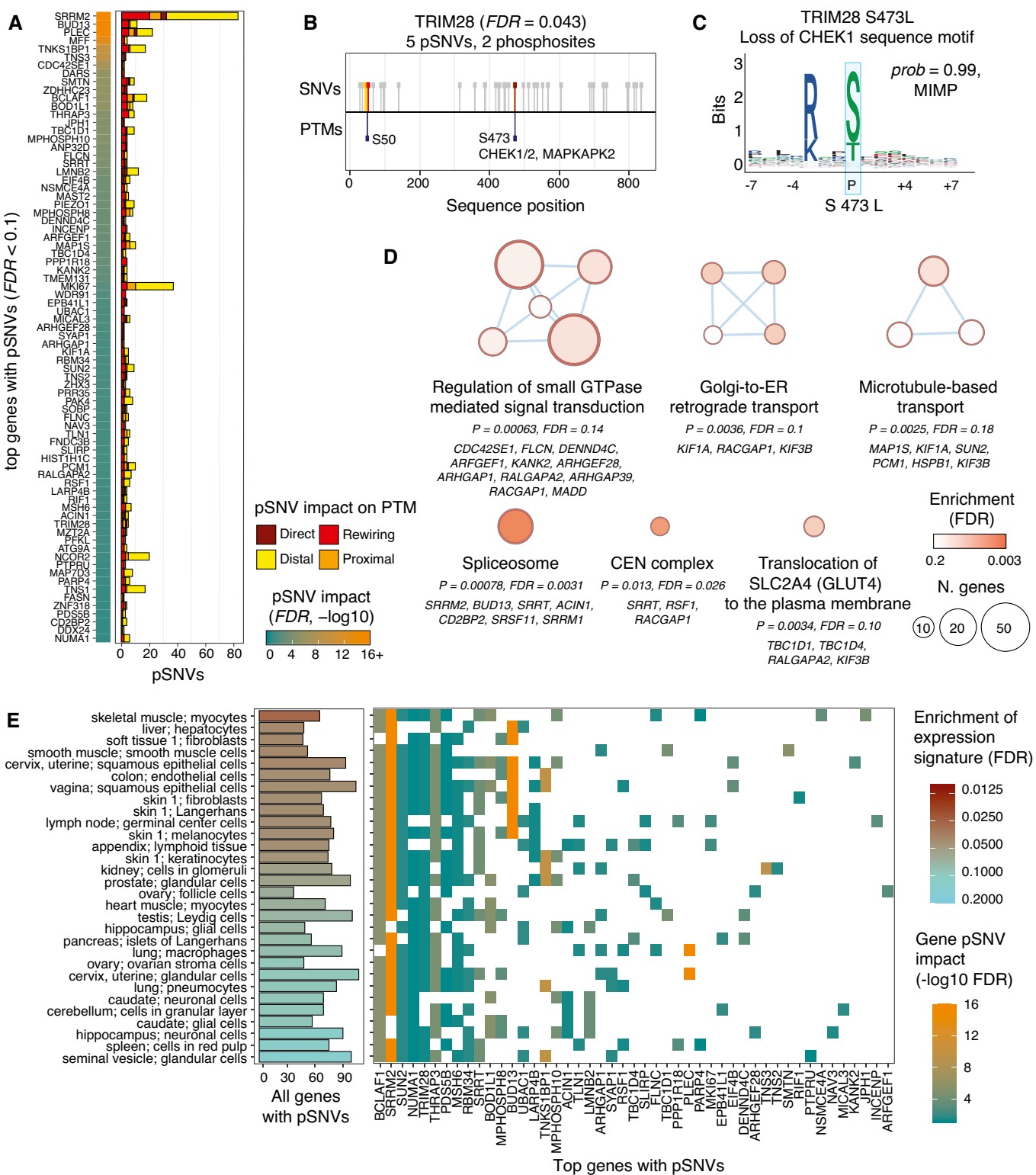

**Figure 2.**

pathways, and protein complexes. Pathway enrichment analysis highlighted 15 gene sets with frequent pSNVs (*FDR* < 0.2 from ActivePathways (Paczkowska *et al*, 2020)) (Fig 2D, Dataset EV4). Ras/ Rho Small GTPase (Ras/Rho) signal transduction was the largest functional theme with eight top genes (*CDC42SE1, FLCN, DENND4C, ARFGEF1, KANK2, ARHGEF28, ARHGAP1, RALGAPA2*) and 39 genes

with pSNVs in total (*FDR* = 0.14, GO:0051056). Ras/Rho signaling networks respond to extracellular stimuli to activate diverse cellular pathways such as proliferation, migration, apoptosis, and cell adhesion. Multiple endoplasmic reticulum (ER) and microtubule-based transport processes were also found. For example, the Reactome pathway *COPI-dependent Golgi-to-ER retrograde traffic* (*FDR* = 0.10, REAC:R-HSA-6811434) with eight enriched genes included the kinesins *KIF1A* and *KIF3B* involved in intracellular transport, and the GTPase signaling protein *RACGAP1*. SARS-CoV-2 uses the host translational machinery in the ER for replication, resulting in increased ER stress and activation of the unfolded protein response pathway (Knoops *et al*, 2008; Ostaszewski *et al*, 2021). We also found the spliceosome protein complex with five top genes (*SRRM2, BUD13, SRRT, ACIN1, CD2BP2*) and 17 genes with pSNVs in total (*FDR* = 0.0031, CORUM:351), extending our observations from the top gene list. Pathway analysis also captured additional genes with pSNVs that remained undetected in the gene-focused analysis. Pathway analysis shows that pSNVs occur in multifunctional genes involved in host processes that are important for the virus life cycle.

Genes with frequent pSNVs were expressed in 30 diverse human tissues and cell types according to expression signatures of the Human Protein Atlas (Uhlen *et al*, 2015) (ActivePathways *FDR* < 0.2) (Fig 2E). The genes were often highly expressed in lung pneumocytes and macrophages, confirming their relevance to respiratory tissues infected by SARS-CoV-2, as well as lymph nodes and spleen, which are damaged by SARS-CoV-2 infection (preprint: Feng *et al*, 2020). Brain and nervous system tissues were also identified, in line with broad cellular perturbations of brain tissues in severe COVID-19 (Yang *et al*, 2021). Gene expression signatures of skin, kidney, pancreas, colon, liver, and female reproductive tissues were also found. These enrichments were partially driven by genes with high expression in most tissues, such as those involved in RNA splicing (*SRRM2, BCLAF1*), interferon regulation (*TRIM28*), cell cycle (*NUMA1, SUN2*), and DNA damage response (*MSH6*). SARS-CoV-2 affects diverse human tissues directly through the ACE2 receptor and indirectly through inflammation and immune response dysregulation, causing broad organ damage in severe disease (Gupta *et al*, 2020). These pathway and tissue-based enrichment analyses were performed relative to the stringent background set of SARS-CoV-2-associated human phosphoproteins, resulting in attenuated statistical power due to multiple-testing corrections. In summary, pathway annotations and expression patterns of genes with pSNVs suggest that these multifunctional genes and pSNVs may contribute to the interindividual variation of SARS-CoV-2 infection and COVID-19.

## Protein-protein interaction networks with frequent SARS-CoV-2-associated pSNVs

We analyzed protein–protein interactions (PPI) to understand the functional context of the top genes with pSNVs (Fig 3A, Dataset EV5). Using the BioGRID database (Oughtred *et al*, 2019), we included 30 physical PPIs among the 77 top human proteins, as well as their 139 interactions with SARS-CoV-2 proteins. We also included 29 high-confidence kinase-substrate interactions with specific SARS-CoV-2-associated phosphosites from the ActiveDriverDB database (Krassowski *et al*, 2021). Kinase–substrate interactions were limited to phosphosites with at least one pSNV. To evaluate the significance of this PPI network, we performed degree-controlled sampling of SARS-CoV-2 phosphoproteins to generate randomized PPI networks as controls.

Top human proteins with pSNVs interacted with viral proteins significantly more often than expected from the human phosphoproteome (139 observed PPI *vs.* 92 $\pm$ 9 expected ($\pm$ 1 SD); permutation $P = 10^{-4}$) (Fig 3B). In contrast, PPIs among the top proteins were less frequent than expected from the control networks (30 versus 52 $\pm$ 7 expected; $P = 3.0 \times 10^{-4}$). Kinase-substrate interactions with pSNVs were over-represented at a statistically sub-significant level compared to control networks (29 observed versus 23 $\pm$ 5 expected; $P = 0.12$).

In the virus-host interactome, the most frequently interacting human proteins (DARS, TMEM131, ARHGAP1, SUN2) and viral proteins (M, nsp4, nsp6, orf7a, orf7b) each had at least 10 PPIs. SUN2 with 9 pSNVs (*FDR* = 0.021) encodes a nuclear membrane protein whose over-expression blocks HIV-1 replication and induces changes in nuclear shape (Popkin *et al*, 2020). RNA-binding proteins and spliceosome subunits such as BUD13, DDX24, and RBM34 interact with orf14, a currently uncharacterized SARS-CoV-2 protein (Zhang *et al*, 2018; Baruah *et al*, 2020). DDX24 is involved in RNA packaging of HIV into virions (Ma *et al*, 2008). RBM34 has been implicated in Middle Eastern Respiratory Syndrome (MERS) infections via interactions with the viral protein nsp3.2 (preprint: Almasy *et al*, 2021). The SARS-CoV-2 non-structural proteins nsp4 and nsp6 are involved in autophagy and viral replication (Cottam *et al*, 2011; Sakai *et al*, 2017) whereas nsp6 suppresses interferon regulatory factor 3 (IRF3) phosphorylation by binding TANK binding kinase 1 (TBK1) (Xia *et al*, 2020). Thus, the human proteins with frequent pSNVs are involved in infection and pathogenesis through interactions with viral proteins, suggesting that some pSNVs could impact SARS-CoV-2 virus-host protein interactions and signaling pathways.

## pSNVs in TBC1D4 associate with glucose homeostasis

We examined the site-specific kinase-substrate interaction network for additional details of pSNVs. The enriched pathway *Translocation of SLC2A4 (GLUT4) to the plasma membrane* (Reactome: HSA-1445148) identified in our pathway analysis is also apparent in the PPI network and includes pSNVs in TBC1D1, TBC1D4, RALGAPA2, KIF3B and others. TBC1D4 (*i.e.*, AS160) is a GTPase-activating protein that regulates glucose homeostasis through insulin-dependent trafficking of GLUT4 (Sano *et al*, 2003). Protein-coding SNVs in TBC1D4 and TBC1D1 have been associated with insulin resistance, obesity, and type II diabetes (Stone *et al*, 2006; Moltke *et al*, 2014). Type II diabetes and obesity are SARS-CoV-2 comorbidities and may implicate certain individuals and populations as more susceptible to severe COVID-19 (Popkin *et al*, 2020).

TBC1D4 includes three pSNVs that alter SARS-CoV-2-associated phosphosites S318 and S588 (*FDR* = 0.0011). TBC1D4 phosphorylation at S588 regulates GLUT4 translocation and subsequent activity in adipocytes (Sano *et al*, 2003). This phosphosite is bound by AKT1, SGK1, RPS6KA1, and CHEK1 kinases based on earlier experimental studies (Geraghty *et al*, 2007; Blasius *et al*, 2011; Coffey *et al*, 2015) (Fig 3C). In our predictions, the pSNV R585Q disrupts sequence motifs at S588, including motifs of CHEK1, SGK1, and RPS6KA1 (Fig 3D). Therefore, the kinases known to phosphorylate the site match the motifs disrupted by the pSNV, indicating that the signaling of TBC1D4 is altered in some individuals due to genetic

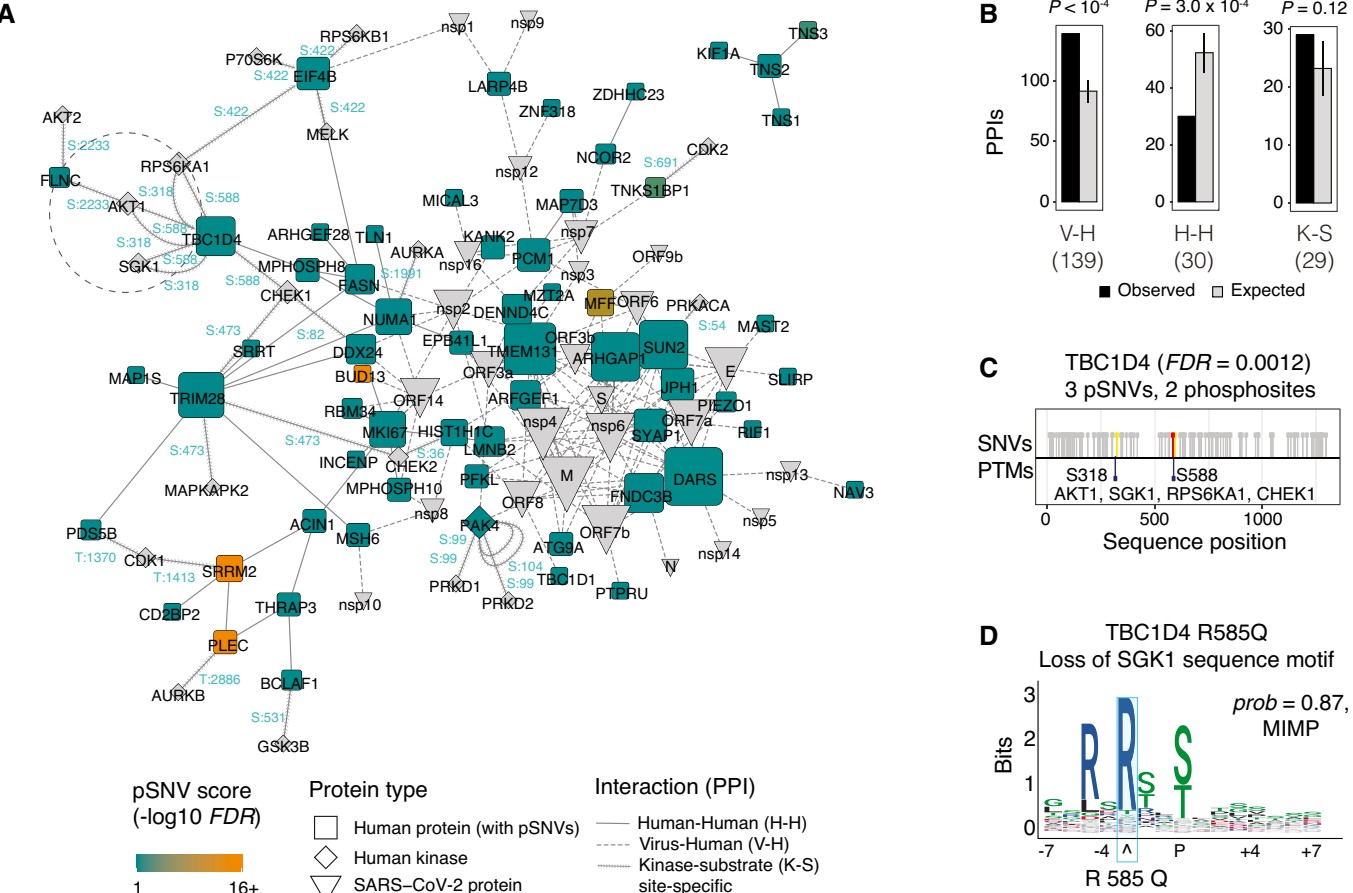

**Figure 3. Protein-protein interactions (PPI) of genes with pSNVs.**

A  The PPI network includes three classes of proteins: (i) human proteins encoded by top genes with pSNVs (*FDR* < 0.1; blue and orange squares), (ii) proteins encoded by the SARS-CoV-2 virus (grey wedges), and (iii) human kinases (grey diamonds) binding SARS-CoV-2-associated phosphosites with pSNVs. Three types of interactions are shown: PPIs of the top human proteins (H-H; solid line), PPIs of the top human proteins and SARS-CoV-2 proteins (V-H; dashed line), and kinase-substrate interactions at phosphosites with pSNVs (K-S; arrow line). Phosphosites with pSNVs labeled in blue. Node size corresponds to the number PPIs per protein.

B  Enrichment analysis of PPIs. Expected PPI counts were derived from control PPI networks sampled randomly using a degree-controlled method on SARS-CoV-2-associated human phosphoproteins. *P*-values of permutation tests are shown (10,000 permutations). Error bars represent ± 1 SD.

C  Example of kinase-substrate interactions altered by pSNVs (dotted circle in panel A). The GTPase signaling protein TBC1D4 involved in glucose homeostasis and transport, diabetes, and obesity, includes three pSNVs in the phosphosites S318 and S588. The sites are phosphorylated by the kinases AKT1, CHEK1, RPS6KA1, and SGK1.

D  The pSNV R585Q in TBC1D4 disrupts the sequence motif of the SGK1 kinase, potentially disrupting glucose homeostasis regulated by TBC1D4.

variation. Besides TBC1D4, the related protein TBC1D1 has nine pSNVs in the phosphosites S614, S627, and S660, including four pSNVs potentially disrupting phosphorylation (*FDR* = $3.8 \times 10^{-5}$); however, the kinases acting on these sites are unknown.

We found that the phosphorylation sites modified in SARS-CoV-2 infection also include phosphosites in a molecular pathway involved in human physiology and diseases that have been implicated as comorbidities of COVID-19. Furthermore, certain pSNVs in these phosphosites have functional predictions of altering pathway activity, while other SNVs in the genes have been associated with the same disease phenotypes in genetic studies. However, our observations should be interpreted with caution since the data are not sufficient to infer a causal relationship of pSNVs, diabetes or obesity, and COVID-19. In summary, some pSNVs in our analysis provide mechanistic and epidemiological hypotheses for studying the interactions of COVID-19 and human phenotypes and diseases.

**Kinase-substrate networks with frequent mutations in sequence motifs**

To characterize the pSNVs in SARS-CoV-2-responsive signaling networks, we studied the kinase families whose sequence motifs were most altered by pSNVs. We computed significance scores for kinase-substrate pairs across all high-confidence motif predictions, giving a higher priority to pairs where multiple pSNVs occurred in the same type of motif. Collectively, we included 217 pSNVs covering 64 kinase families. Since the individual pSNVs were often associated with several similar sequence motifs of related kinases in MIMP predictions, this analysis combined 3,360 high-confidence motif-rewiring predictions across the pSNVs and resulted in partially redundant functional impact scores.

Hierarchical clustering of motif-rewiring pSNVs in top genes revealed two major clusters (Fig 4A). The first cluster included

pSNVs disrupting motifs of cyclin-dependent kinases (CDKs) and mitogen-associated kinases (MAPK). For instance, CDK7 motifs were lost through 56 pSNVs and gained through two pSNVs (Fig 4B). The proliferation marker MKI67, the DNA damage response protein MSH6, the spliceosome subunit SRRT, the interferon regulator BCLAF1, and the translation initiation factor EIF4B included such pSNVs. SARS-CoV-2 infection is known to promote activation of host MAPK p38 kinases and inhibition of CDKs (Bouhaddou et al, 2020).

The second, larger cluster involved motif alterations of Protein Kinase A, CAMK, checkpoint, and PI3K/Akt kinases. The PI3K/Akt pathway controls cell proliferation, survival, and suppression of apoptosis, and responds to extracellular signals of cytokines and growth factors via the mTOR pathways. Checkpoint kinases respond to DNA damage by blocking cell proliferation and inducing cell death. Among the top proteins, motifs bound by Protein Kinase A (PRKACA) and CAMK2A were altered by 98 pSNVs, most of which led to motif losses or replacements of central phospho-residues (Fig 4B). Motif-rewiring pSNVs occurred in transcriptional and post-transcriptional regulators such as PLEC, BUD13, THRAP3, and TRIM28. Therefore, pSNVs may reconfigure kinase signaling networks of defense processes and host components of the viral life cycle and contribute to disease variation.

### Motif-switching pSNVs exchange CDK and MAPK kinase targets in favor of TBK1

We focused on the subset of motif-switching pSNVs that disrupted one type of sequence motif and induced another type of motif at the same mutated phosphosite, potentially causing switches in kinase binding. This revealed a network of 54 proteins with pSNV-induced binding switches involving 26 kinase families, when selecting the most high-scoring kinase family for every pSNV (Fig 4C, Dataset EV6). Motif switches were caused by 60 pSNVs, significantly more often than expected from phosphosites sampled from the human proteome ($26 \pm 10$ expected, permutation $P < 10^{-4}$) (Fig 4D).

The largest subnetwork of motif switches involved motif gains of Tank Binding Kinase 1 (TBK1) of the IKK family (26 pSNVs) that replaced motifs of other kinases, predominantly CDKs and MAPKs (23 pSNVs observed, $10 \pm 3$ expected, $P = 1.0 \times 10^{-4}$) (Fig 4E). TBK1 was also the top kinase with most motif gains overall (37 pSNVs observed, $18 \pm 4$ expected, $P = 1.0 \times 10^{-4}$). The transcriptional regulators BCLAF1 and RREB1, the WDR91 protein involved

in host cell entry of SARS-CoV-2 (Zhu et al, 2021), the nuclear matrix protein NUMA, and the proliferation marker MKI67 included such motif-switching pSNVs. TBK1 controls immune monitoring pathways and phosphorylates interferon regulatory factors IRF3 and IRF7 to trigger the host antiviral response that is suppressed by SARS-CoV-2 proteins (Sharma et al, 2003; Balka et al, 2020; Lei et al, 2020; Xia et al, 2020). Thus, in some individuals and populations, motif-rewiring pSNVs may induce structural changes in kinase signaling networks and crosstalk of mitogenic and immune response pathways. These network-rewiring pSNVs potentially reflect the inter-individual heterogeneity and evolutionary pressures of signaling networks responding to viral infections.

### Associations of pSNVs with severe COVID-19 and hospitalization

We asked if pSNVs associated with COVID-19 phenotypes in the Regeneron meta-study of more than 500,000 individuals covering seven COVID-19 outcomes and more than 8,000 individuals with COVID-19 (Kosmicki et al, 2021a). We focused on the pan-ancestry exome-sequencing dataset that combined multiple cohorts (Geisinger, UK Biobank, UPENN) and included a fraction of the pSNVs we identified (760 or 37%). To select significant associations, we used a restricted FDR approach that only considered pSNVs in multiple testing correction for increased sensitivity.

We found ten pSNVs that associated with COVID-19 phenotypes (restricted $FDR < 0.1$) (Fig 5A, Dataset EV7). Eight pSNVs associated with severe COVID-19 compared to a broad control group of COVID-19-negative or COVID-19-unknown individuals. The pSNV R232H in ARHGEF2 showed the strongest association ($FDR = 2.9 \times 10^{-4}$, $P = 4.3 \times 10^{-7}$, odds ratio (OR) = 210, 95% confidence interval (CI) = 26–1,673). ARFGEF2 regulates membrane trafficking between the trans-Golgi network and endosomes (Ishizaki et al, 2008). The overall allele frequency of this risk-associated pSNV was low ($AF_{all} = 7.2 \times 10^{-5}$) and it was the highest in the European population ($AF_{nfe} = 1.2 \times 10^{-4}$) (Fig 5B). Other ARFGEF2 mutations have been linked to autosomal recessive periventricular heterotopia with microcephaly (ARPHM), a rare disorder involving cerebral malformations, severe developmental delay, and recurrent pulmonary infections, whereas loss-of-function experiments implicated the gene in neuronal proliferation and migration (Sheen et al, 2004). The R232H pSNV is indicated as a variant of unknown significance for ARPHM in the ClinVar database (see below). Seven additional infrequent pSNVs associating with severe COVID-19 were

---

**Figure 4. Kinases involved in sequence motif rewiring through pSNVs.**

A   Heatmap shows the hierarchical clustering of top genes with pSNVs (X-axis) and kinase families with sequence motif rewiring (Y-axis). Genes and kinases are quantified by scores that reflect the impact of pSNVs on motifs or phosphoresidues. Bar plots show pSNV counts in sequence motifs (left) and top genes (bottom). Counts are redundant since pSNVs often alter multiple similar motifs. Scores are capped at 5 for visualization.

B   Examples of sequence motifs rewired by pSNVs: cyclin-dependent kinase 7 (CDK7) (left), protein kinase CAMP-activated catalytic subunit alpha (PRKACA) (right). Sequence logo of the motif (top) and bar plot with pSNVs altering the motif (bottom) are shown.

C   Network of pSNVs causing motif switches (i.e., combined gain of one motif and loss of another motif). Nodes show kinase families (diamonds) and top genes (circles), and edges show motif losses (red) and gains (green) caused by pSNVs. Node size indicates the number of gained and lost motifs. For each pSNV, only the kinase family with highest-scoring motif is shown.

D   Enrichment analysis of motif-switching pSNVs. Expected values were derived by sampling equal numbers of phosphosites from the human phosphoproteome. P-values of permutation tests are shown (10,000 permutations). Error bars show $\pm 1$ SD.

E   Sequence motifs bound by the TANK binding kinase 1 (TBK1) of the IKK family are gained through pSNVs, while motifs bound by CDK and MAPK kinase families are lost. Bar plots show the numbers of pSNVs causing gains of TBK1 motifs (top) and losses of MAPK14 motifs (bottom). A subset of these pSNVs cause motif switches (panels C and D). pSNVs in top genes are included in panels A and B, and pSNVs in all genes are included in panels C-E.

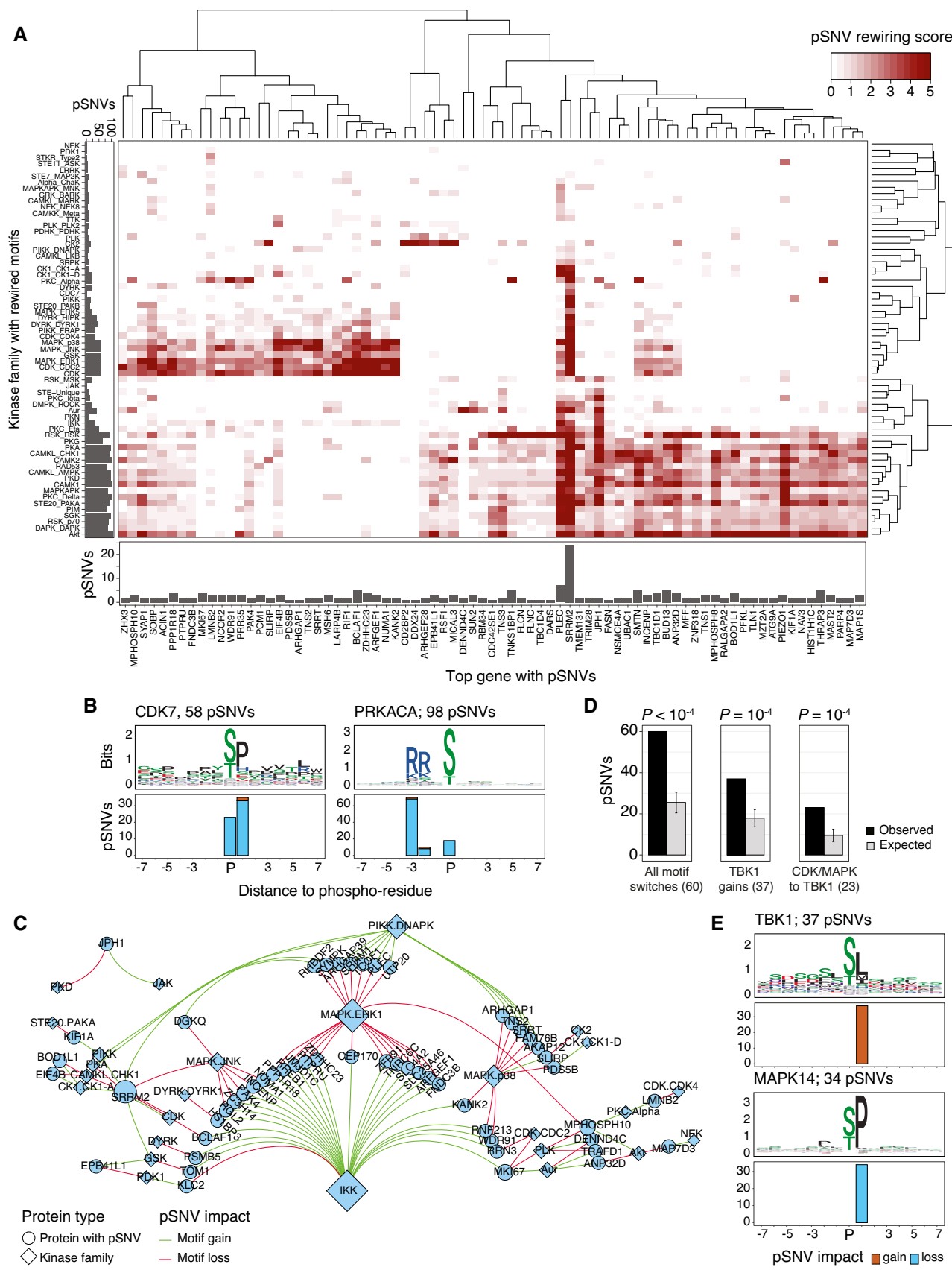

**Figure 4.**

found in other genes (*SLC4A2*, *CENPF*, *MPHOSPH8*, *ZC3HC1*, *PRR35*, *PSMF1*, *EPS15L1*).

The pSNV R383P in STARD13 negatively associated with COVID-19 hospitalization compared to COVID-19 positive non-hospitalized individuals, suggesting a small protective effect (*FDR* = 0.087, *P* = 0.0016, OR = 0.64, 95% CI = 0.49–0.84). The pSNV is relatively frequent in the human genome, with 1.9% overall frequency in gnomAD and up to 2.9% frequency in northern European populations. STARD13 is a Rho GTPase signaling protein and a tumor suppressor in liver cancer (Ching *et al*, 2003). In pancreatic islets, the protein regulates insulin secretion and actin cytoskeleton remodeling in response to glucose (Naumann *et al*, 2018), highlighting another potential functional link with COVID-19 comorbidities.

To further study these risk associations, we examined the GWAS dataset of the COVID-19 Host Genetics Initiative that covered nearly 50,000 COVID-19 patients using genotyping and imputation (Covid-19 Host Genetics Initiative, 2021a). The pSNV in STARD13 also negatively associated with hospitalized versus non-hospitalized COVID-19 individuals (unadjusted *P* = 0.033, log OR = −0.11 ± 0.052 SE) (Fig 5C, Dataset EV8). Also, a negative association of this pSNV with severe respiratory disease and hospitalization compared to the overall population was apparent at sub-significant levels (unadjusted *P* ≤ 0.12). The other pSNVs we identified were not reported in the GWAS, likely due to their low allele frequency. The twice-identified association of the pSNV in STARD13 suggests that it may have weak protective effects against severe COVID-19. However, these results need to be interpreted with caution due to their limited statistical significance.

### Population genetic variation, positive selection, and disease associations of pSNVs

We studied the allele frequencies of pSNVs in gnomAD to understand the extent of genetic variation in SARS-CoV-2 responsive signaling networks (Fig 5D). Most pSNVs were relatively infrequent overall (median $AF_{all} = 3.8 \times 10^{-5}$). However, the allele frequencies of pSNVs in their most representative populations were an order of magnitude higher (median $AF_{popmax} = 2.1 \times 10^{-4}$), suggesting the value of deeper population-based analyses of pSNVs.

We found 86 relatively frequent pSNVs ($AF_{popmax} > 1\%$), including 15 pSNVs causing sequence motif rewiring or loss of phospho-residues. For example, the common pSNV P257L in WDR91 ($AF_{all} = 0.72$) altered the phosphosite S256 by inducing TBK1 motifs and disrupting CDK and MAPK motifs. *WDR91* encodes a WD repeat-containing protein that was identified as a host regulator of endosomal entry of SARS-CoV-2 in a CRISPR screen (Zhu *et al*, 2021). Frequent motif-rewiring pSNVs were also found in the

spliceosome subunit *SRRM2*, the pro-apoptotic splicing and transcription factor *BCLAF1*, and the HIV-related nuclear membrane protein *SUN2*.

Variation in allele frequency was also apparent for the genes and pathways we prioritized. For example, the pSNV S473L in TRIM28 was exclusively found in the African population ($AF_{afr} = 1.2 \times 10^{-4}$). The adjacent pSNV R472C detected in admixed American, northwestern European, and other populations ($10^{-5} < AF < 10^{-4}$) may also contribute to altered signaling; however, no motif disruptions were identified for this pSNV. As another example, the pSNV TBC1D4 R585Q was detected in south Asian, Latino, and north-western European populations ($AF_{sas} = 1.3 \times 10^{-4}$; $AF_{amr} = 2.9 \times 10^{-5}$; $AF_{nfe\_nwe} = 4.8 \times 10^{-5}$). Additional pSNVs in TBC1D1 and TBC1D4 were found in other populations and may also contribute to network rewiring in SARS-CoV-2 infection.

We found 86 pSNVs associated with human disease in the ClinVar database (Landrum *et al*, 2020) (Dataset EV9), including cancer predisposition (*NF1*, *PMS2*, *FLCN*, *MSH6*, *APC*), cardiovascular phenotypes (*BMPR2*, *MYLK*, *VCL*, *LMNA*), muscular dystrophy (*SUN2*, *FLNC*), natural killer cell and glucocorticoid deficiency (*MCM4*), autoimmune interstitial lung, joint, kidney disease (*COPA*), and renal cysts and diabetes syndrome (*HNF1B*). These annotations suggest that SARS-CoV-2 and COVID-19 may hijack disease-related signaling networks. However, since these pSNVs were predominantly annotated as variants of unknown significance (VUS), we have currently no evidence of their involvement in pathogenesis.

Finally, we asked if the genes with pSNVs showed signs of positive selection in humans by analyzing a catalogue of selective sweeps (Johnson & Voight, 2018) in the 1000 Genomes project. Positive selection in at least one human population was apparent in 215 genes with pSNVs (23%), including 23/77 of the top genes (Fisher's exact *P* = 0.078) as well three genes with COVID-19 risk associations (Fig 5E, Dataset EV10). This evidence of evolutionary selection complements our earlier observation of enriched motif-rewiring pSNVs in SARS-CoV-2-associated signaling networks.

## Discussion

We highlighted human genetic variation that may rewire phospho-signaling networks responding to SARS-CoV-2 infection, using a machine-learning approach that identified impactful mutations in protein sequence motifs. Motif-rewiring pSNVs were found in host processes of the viral life cycle such as RNA splicing, innate immune responses such as interferon activation, Ras/Rho GTPase signaling proteins, and in genes involved in virus infections. Genes with

---

**Figure 5. COVID-19 risk associations and evolutionary selection of pSNVs.**

A pSNVs with COVID-19 risk associations from an exome-sequencing study of COVID-19 outcomes (Kosmicki *et al*, 2021a). Volcano plot highlights ten pSNVs associated with COVID-19 phenotypes at a restricted significance threshold (*FDR* < 0.1; only pSNVs tested). Dot size corresponds to maximum allele frequency among populations ($AF_{popmax}$).

B Allele frequencies of COVID-19-associated pSNVs in gnomAD.

C Risk association of the pSNV in STARD13 in the GWAS of the COVID-19 Host Genetics Initiative (Covid-19 Host Genetics Initiative, 2021a).

D Histogram of $AF_{popmax}$ of all pSNVs in gnomAD grouped by functional impact. High-frequency pSNVs ($AF_{popmax} \geq 1\%$) with direct or motif-rewiring impact are listed.

E Integrated haplotype scores of genes with pSNVs reflecting positive selection (Johnson & Voight, 2018). Top genes with pSNVs are labeled. Asterisks show genes with COVID-19 risk associations.

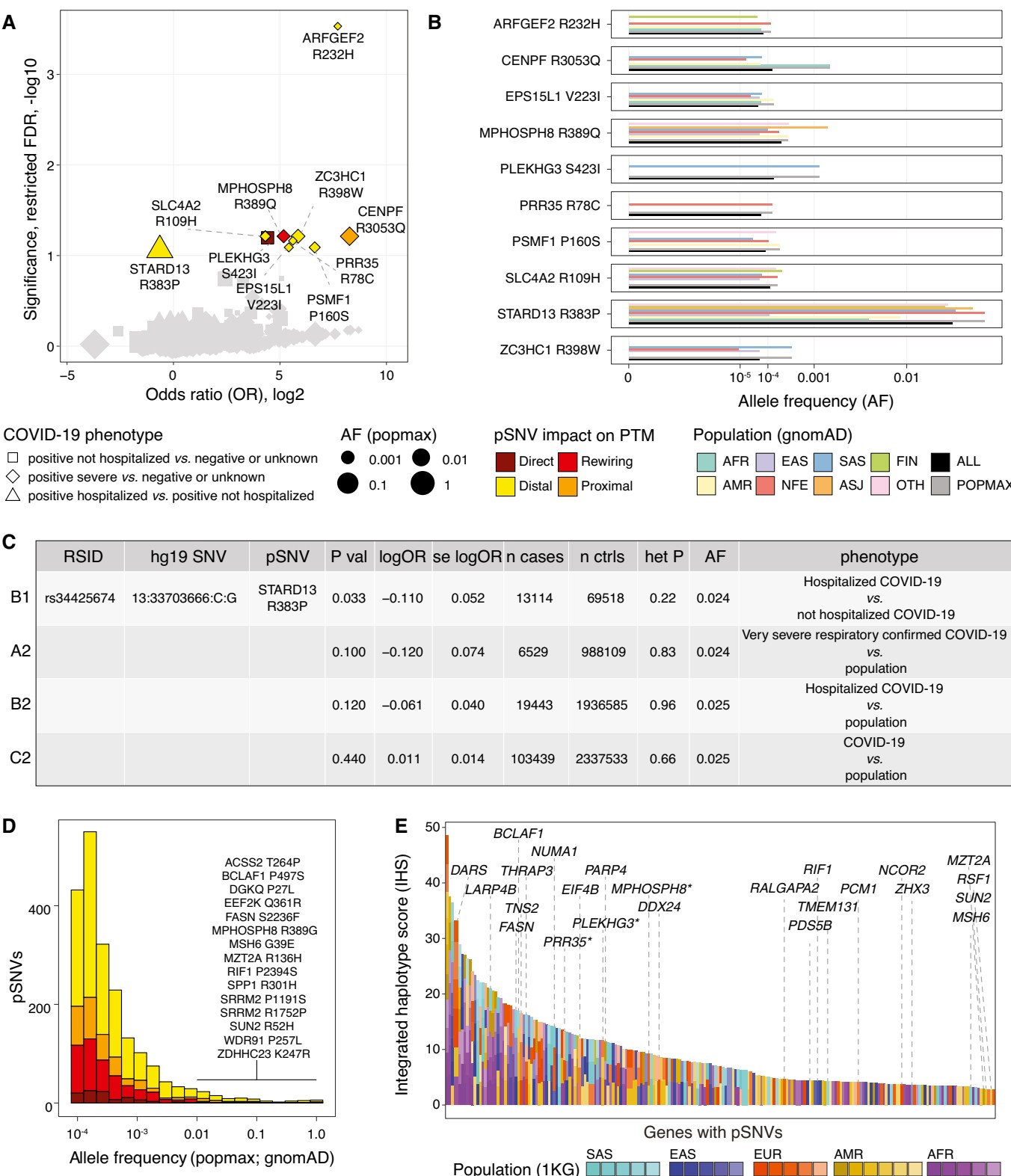

**Figure 5.**

pSNVs were often expressed in diverse human tissues, reflecting the broad organotropism of SARS-CoV-2 and potential contribution of pSNVs to multi-organ failure in severe COVID-19, as well as its long-term symptoms. Disease genes and variants of unknown significance found among pSNVs suggest that infections may hijack disease pathways. Lastly, several pSNVs associated with COVID-19 phenotypes,

including one variant with a potentially protective effect against severe disease that we confirmed in an additional cohort.

The proteome-wide enrichment of motif-rewiring pSNVs suggests evolutionary plasticity of the human signaling network responding to virus infection. This evolution of host defense strategies is exemplified by the number of motif-switching pSNVs that consistently induce novel sequence motifs of TBK1 and simultaneously disrupt motifs recognized by MAPK and CDK kinases. On the one hand, a quarter of pSNVs are in positively selected genomic regions. On the other hand, human phosphorylation sites are generally depleted of genetic variation, indicating an increased evolutionary constraint (Reimand *et al*, 2015). Therefore, the enrichment of pSNVs with network-rewiring effects in SARS-CoV-2 infection may correspond to positive evolutionary pressures of a host-pathogen arms race as well as relaxed selection in the overall context of conserved signaling networks.

Our study has important limitations. First, we have only little evidence of pSNVs associating with SARS-CoV-2 infection, COVID-19 risk or comorbidities, as most pSNVs are infrequent and challenging to link with clinical phenotypes. Our analysis captures risk variants by focusing on a smaller set of potentially functional pSNVs and limiting the effects of multiple testing correction, thus revealing subtler associations that are missed in genome-wide studies. Also, as these phosphosites reflect an early post-infection timepoint in cell culture, the coverage of signaling pathways activated further downstream of SARS-CoV-2 infection in human tissues and the immune system is limited. Our permutation analysis of pSNV impact considers sites from the entire human phosphoproteome as controls, and thus also emphasizes features of pSNVs that are characteristic to the SARS-CoV-2-responsive signaling networks. On the one hand, we may overestimate the extent of network rewiring because our pSNV analysis does not account for co-expression and localization of kinases and substrates. On the other hand, the number of motif-rewiring pSNVs may be underestimated, because our knowledge of sequence motifs is incomplete. The landscape of pSNVs may be extended to other types of PTMs.

Our integrative proteogenomic analysis of pSNVs in SARS-CoV-2 responsive signaling networks enables further studies of viral infection, disease mechanisms, and translation. Deeper analysis of genome sequencing datasets with matched COVID-19 profiles is needed to associate pSNVs with patient risk and co-morbidities and enable biomarker development. The candidate genes, pathways, and kinases we identified enable functional genomics screens and phenotypic experiments that may improve mechanistic understanding of SARS-CoV-2 infection and COVID-19. The processes and signaling networks involve known drug targets, some of which have already been proposed in recent studies. Thus, our findings can contribute to genome-guided risk predictions and therapies for the current and future pandemics.

# Materials and Methods

## Phosphorylation sites in human proteins

Protein phosphosites differentially phosphorylated in SARS-CoV-2 infection were retrieved from the phosphoproteomics study by Bouhaddou *et al* (Bouhaddou *et al*, 2020). We mapped the phosphosites to canonical protein isoforms (hg19) using the Active-DriverDB database (version 2021) (Krassowski *et al*, 2021; Data ref: the ActiveDriverDB database, 2021). Sites with phosphorylation increases and decreases were included. SARS-CoV-2-associated phosphosites detected at the 24 h timepoint were filtered for significance (*FDR* < 0.05 in infected cells; *FDR* > 0.05 in controls). We removed sites corresponding to non-phosphorylated residues (*i.e.*, other than S, T, or Y). In total, we analyzed 1,530 SARS-CoV-2-associated phosphosites in 949 proteins. To compare observed and expected numbers of pSNVs, all experimentally detected human phosphosites were retrieved from ActiveDriverDB based on proteomics databases (PhosphoSitePlus (Hornbeck *et al*, 2015), UniProt (UniProt Consortium, 2019), Phospho.ELM (Dinkel *et al*, 2011), HPRD (Keshava Prasad *et al*, 2009)). SARS-CoV-2-associated phosphosites were excluded from these controls. Known interactions of kinases and specific phosphosites were also retrieved from ActiveDriverDB.

## Human genome variation

Genome variation maps of human populations of 124,748 individuals were retrieved from the gnomAD exome sequencing project (version 2.1.1; hg19) (Karczewski *et al*, 2020a; Data ref: Karczewski *et al*, 2020b). Single nucleotide variants (SNVs) were first filtered by variant quality (*i.e.*, with *filter = PASS*) and maximum allele frequency per population ($AF_{popmax} \geq 10^{-4}$). The ANNOVAR software (Wang *et al*, 2010) (version 2019-10-24) was used to annotate the SNVs in protein-coding genes. Missense SNVs in canonical protein isoforms were selected in order to match the protein phosphosites in ActiveDriverDB. We excluded small insertions-deletions (indels), splicing, frameshift, and stop codon mutations, and missense SNVs with mismatching reference amino acid residues.

## pSNV impact prediction

Phosphorylation associated SNVs (pSNVs) occurred in SARS-CoV-2-associated phosphosites in flanking sequences of ± 7 residues. Four mutually exclusive categories of variant impact were assigned in order of priority: direct (substitution of phosphorylated residues S, T, Y), motif-rewiring (gain or loss of motif), and proximal and distal (within 1-2 and 3-7 residues from phosphosite, respectively, with no motif-rewiring prediction). In the cases where multiple adjacent phosphosites were found to match a pSNV, the phosphosite of highest impact was used for annotation. Motif-rewiring predictions were derived using the MIMP method (Wagih *et al*, 2015) (v1.2), with the database of high-confidence kinase position weight matrices (PWMs) (*model.data = hconf*), a posterior probability cutoff (*prob > 0.5*), and inclusion of central residues (*include.cent = TRUE*). To score each pSNV, the kinase motif (*i.e.*, the PWM) with the largest posterior probability was selected. To prioritize pSNVs causing motif switches, both posterior probabilities (*i.e.*, the top motif gain and loss) were selected and aggregated as $prob_{switch} = 1 - (1 - prob_{gain})(1 - prob_{loss})$, corresponding to the combined probability of the pSNV causing either a gain or a loss of a motif. Functional scores of human phosphosites were retrieved from an earlier study (Ochoa *et al*, 2020) and merged with our datasets using UniProt IDs and residue positions. We compared the functional scores of SARS-CoV-2-associated phosphosites

with pSNVs with other phosphosites in the human proteome using the non-parametric Wilcoxon rank-sum test.

## Enrichment of motif-rewiring pSNVs

Custom permutation tests were used to evaluate the expected numbers of pSNVs with functional impact predictions (direct, motif-rewiring, proximal, distal) and other pSNV annotations. As controls, we randomly sampled 10,000 sets of phosphosites from the human phosphoproteome such that each sample contained the same number of phosphosites (1,530) as was detected in the SARS-CoV-2 infection experiment. The control phosphosites in the human proteome were retrieved from ActiveDriverDB and excluded the SARS-CoV-2-associated phosphosites. The control sites, from which control pSNVs were derived, were annotated using gnomAD SNVs and MIMP analysis identically to the SARS-CoV-2-associated phosphosites analyzed elsewhere in the manuscript, and control pSNVs (cpSNVs) were derived. The impact of cpSNVs in control phosphosites, their motif-rewiring annotations, and overlap with known kinase binding sites were also derived identically. To analyze the statistical significance of pSNV impact in SARS-CoV-2-associated human phosphosites (*i.e.*, direct, distal, proximal, motif-rewiring), we counted how many times the cpSNVs in the control phosphosites were observed with these impacts across 10,000 iterations. Two empirical *P*-values were computed as the fractions of times the randomly sampled phosphosites produced greater or smaller counts of cpSNVs with similar impact (*i.e.*, $P = n / 10,000$), and the smaller of the two *P*-values was reported as the final *P*-value reflecting a one-tailed test to measure either the over- or under-representation of impact annotations. The expected counts of cpSNVs with specific impacts were derived from the randomly sampled phosphosites and were shown as mean values with $\pm 1$ standard deviation (SD) for confidence intervals. We also used the same strategy and the same randomly sampled sets of control phosphosites to evaluate the significance of additional pSNV annotations over-represented in SARS-CoV-2-associated phosphosites. First, a subset of SARS-CoV-2-associated phosphosites were found to occur in previously annotated binding sites of certain kinases. For each such kinase, we asked how often the cpSNVs in the 10,000 sets of control phosphosites occurred in the known binding sites of the kinase as sampled from the human phosphoproteome. Kinases with over-represented pSNVs in the SARS-CoV-2-associated phosphosites were shown if their frequencies significantly exceeded the expected frequencies of cpSNVs in known kinase binding sites among the 10,000 sets of control phosphosites ($P < 0.05$, permutation test). Second, we counted the pSNVs in SARS-CoV-2-associated phosphosites for which motif-switching impact was predicted (*i.e.*, loss of a sequence motif combined with gain of another motif). To evaluate the enrichment of these pSNVs relative to the human phosphoproteome, we counted the equivalent motif-rewiring cpSNVs in each of the 10,000 sets of control phosphosites sampled from the human phosphoproteome and reported the corresponding *P*-values of permutation tests as described above. Third, we focused on the subset of motif-switching pSNVs in SARS-CoV-2-associated phosphosites that led to losses of CDK/MAPK sequence motifs and simultaneously induced TBK1 motifs. Equivalent counts of CDK/MAPK-to-TBK1 motif-switching cpSNVs were derived from the same 10,000 sets of control phosphosites sampled from the human proteome.

We reported the corresponding *P*-values of permutation tests as described above.

## Gene prioritization

Genes were scored based on the aggregated pSNV impact in phosphosites. Each pSNV was assigned either the predicted posterior probability of motif rewiring ($prob_{SNV} = prob_{rewiring}$ from MIMP; see above), or alternatively, a conservative posterior probability of motif rewiring ($prob_{SNV} = 0.05$) if it was annotated as distal or proximal to the phosphosite with no confident motif-rewiring predictions from MIMP (*i.e.*, for pSNV with $prob < 0.5$). To derive gene-level significance scores (*P*-values), posterior probabilities of all pSNVs in each gene were aggregated as $P_{gene} = 1 - \Pi_{\text{SNV of gene}} (1\text{-}prob_{SNV})$, reflecting the null hypothesis that none of the pSNVs in the protein independently altered kinase binding motifs or central phosphoresidues. The resulting gene-level *P*-values were corrected for multiple testing with the Benjamini–Hochberg false discovery rate (*FDR*) method and genes with significant *FDR* values were selected ($FDR < 0.1$). *FDR* values were capped at $10^{-16}$ for visualization.

## Functional enrichment analysis

Pathway enrichment analysis was conducted using the ActivePathways (Paczkowska *et al*, 2020) method (version 1.1.0) using gene sets of biological processes of Gene Ontology (Ashburner *et al*, 2000), molecular pathways of Reactome (Fabregat *et al*, 2018), protein complexes of CORUM (Ruepp *et al*, 2010), and genes expressed in human tissues of the Human Protein Atlas (Uhlen *et al*, 2015). Gene sets were retrieved from the g:Profiler web server (version 25032021) (Reimand *et al*, 2007; Data ref: the g:Profiler web server, 2021) (Mar 25th, 2021). As statistical background, we used the 949 proteins with phosphosites associated with SARS-CoV-2 infection. ActivePathways compared the *P*-value-ranked list of 693 genes with pSNVs using ranked hypergeometric tests and identified the enrichments of individual gene sets at optimal lengths of the ranked input gene list (Paczkowska *et al*, 2020). All types of gene sets were from independent databases and were therefore analyzed separately (biological processes, pathways, complexes, tissue expression signatures). To reduce the extent of multiple testing in this underpowered analysis, we constrained the input gene sets by size and thus selected gene sets of high functional specificity. For pathways, processes, and protein complexes, we included gene sets with at least 10 and up to 50 infection-specific phosphoproteins. The gene sets of high expression signatures that we analyzed included at least 50 and up to 150 infection-specific phosphoproteins, while low and medium expression signatures were excluded. Significantly enriched pathways were selected in ActivePathways using a relatively lenient false discovery rate threshold to tackle the low power of this analysis ($FDR < 0.2$). Functional gene sets were visualized as an enrichment map (Reimand *et al*, 2019) and the subnetworks were annotated using their major gene sets. To identify further pathway genes with pSNVs that were not found in the gene-focused analysis, leniently filtered genes were shown in the map (gene $FDR < 0.25$). Genes with pSNVs highly expressed in human tissues were visualized separately and their gene-based *FDR* values were capped at $10^{-16}$ for visualization.

### Analysis of protein–protein interaction (PPI) networks

Three types of PPIs were included in the network of top genes with pSNVs. First, we included physical human-human PPIs where both interacting host proteins were part of the top gene list. Second, we included physical human-virus PPIs where one interacting host protein was part of the top gene list, and the other protein was encoded by the SARS-CoV-2 virus. Third, we included experimentally verified site-specific host kinase-substrate interactions where the substrate was part of the top gene list, the kinase was known to bind the substrate phosphosite based on previous studies, and the phosphosite had at least one pSNV. Human–human and human–virus PPIs were retrieved from the BioGRID database (version 4.4.199) (Oughtred *et al*, 2019; Data ref: the BioGRID database, 2021). Kinase-substrate PPIs were retrieved from ActiveDriverDB. The PPI network was visualized using Cytoscape (Cline *et al*, 2007). Degree-weighted PPI networks ($n = 10,000$) were randomly generated as controls to evaluate the expected numbers of the three types of PPIs, using the iGraph R package (version 1.2.11). *P*-values of permutation tests were computed to evaluate the enrichment of PPIs among the top genes with pSNVs. Expected counts were shown as mean values with $\pm 1$ SD for confidence intervals.

### Population frequency, disease annotations, evolutionary selection of pSNVs

Allele frequencies of pSNVs in human populations were reported from gnomAD. Disease associations of pSNVs from the ClinVar database (Landrum *et al*, 2020) mapped in ActiveDriverDB were used. We reviewed these individually on the ClinVar website (Landrum *et al*, 2020; Data ref: the ClinVar database, 2021) (data retrieved on Oct 6th, 2021). Using ClinVar, we filtered low confidence pSNV annotations that lacked star ratings or showed consensus ratings of *benign*. Evolutionary analysis of pSNVs was performed using a catalogue of selective sweeps (Johnson & Voight, 2018) in populations of the 1000 Genomes Project. Top integrated haplotype scores (iHS) for 26 populations were used after excluding the regions with negative standardized iHS. For each gene with pSNVs, we reported the mean iHS value of its pSNVs for each human population. Enrichment of top genes with pSNVs among these genes was evaluated with a one-sided Fisher's exact test.

### COVID-19 risk analysis of pSNVs

COVID-19 risk associations of pSNVs were retrieved from the Regeneron meta-analysis (Kosmicki *et al*, 2021a; Data ref: Kosmicki *et al*, 2021b) of multiple exome-sequencing datasets (Geisinger, UK Biobank, UPENN; *_meta_ghs_ukb_upenn_all_exome_20210322.csv.gz; downloaded on Jan 27th 2022). All associations were merged with our pSNV datasets using RSIDs. For each COVID-19 phenotype, a restricted Benjamini-Hochberg FDR correction was applied such that only the pSNVs that were merged with risk associations via RSIDs were corrected for multiple testing and filtered for significance (*FDR* < 0.1), resulting in a lenient selection threshold. The ten risk-associated pSNVs were further examined in the GWAS dataset of the COVID-19 Host Genetics Initiative (Covid-19 Host Genetics Initiative, 2021a; Data ref: Covid-19 Host Genetics Initiative, 2021b) and one pSNV was available

in the data (STARD13 R383P). We reported unadjusted *P*-values for all four COVID-19 phenotypes for this pSNV.

### Phosphosites of other virus infections

Differentially phosphorylated sites associated with infection of Herpes simplex virus type 1 (HSV-1) (Kulej *et al*, 2017) and human immunodeficiency virus 1 (HIV-1) (Greenwood *et al*, 2016) were also analyzed. Sites with at least 75% localization probability were selected and filtered (*FDR* < 0.05). For HSV-1 infection, phosphorylation changes of the 15-h time point were selected (ANOVA). For HIV infection, phosphorylation changes comparing the infected and mock cells at the 48-h time point were selected based on the analysis with the limma method (Ritchie *et al*, 2015). Sites with phosphorylation increases and decreases were included in both cases. For the sites resolved multiple times and with no likelihood estimates provided in the original study, the resolutions with lower multiplicity and higher *P*-value were selected conservatively. The phosphosites were mapped to protein sequences as described previously (Krassowski *et al*, 2021). We used a two-tailed Fisher's exact test to evaluate the intersection of the SARS-CoV-2-associated phosphosites with pSNVs and the shared SARS-CoV-2-associated phosphosites.

# Data availability

This study includes no data deposited in external repositories.

## Acknowledgements

We would like to thank Lili Milani, Nina Adler, and the reviewers for constructive comments and Mykhaylo Slobodyanyuk for technical assistance. This work was supported by COVID-19 Supplements of the Project Grant from the Canadian Institutes of Health Research (CIHR) to J.R. (PJT-162410). The Discovery Grant from the Natural Sciences and Engineering Research Council (NSERC) to J.R. (RGPIN-2016-06485). This work was supported by the Investigator Award to J.R. from the Ontario Institute for Cancer Research (OICR). A.B. was supported by an Ontario Graduate Scholarship. M.K. was supported by the Scatcherd European Scholarship. Funding to OICR is provided by the Government of Ontario.

## Author contributions

**Diogo Pellegrina:** Data curation; Formal analysis; Visualization; Methodology; Writing—original draft; Writing—review & editing. **Alexander T Bahcheli:** Data curation; Formal analysis; Visualization; Methodology; Writing—original draft; Writing—review & editing. **Michal Krassowski:** Data curation; Formal analysis; Visualization. **Jüri Reimand:** Conceptualization; Data curation; Formal analysis; Supervision; Funding acquisition; Writing—original draft; Writing—review & editing.

In addition to the CRediT author contributions listed above, the contributions in detail are:
DP, AB and JR analyzed the data, interpreted the results, and prepared the figures. MK preprocessed the data and led the database development. DP, AB, and JR wrote the manuscript. JR conceived and supervised the project. All authors reviewed and edited the manuscript and approved the final version.

## Disclosure and competing interests statement

The authors declare that they have no conflict of interest.

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
