## [Review Process File · Molecular Systems Biology]

Human phospho-signaling networks of SARS-CoV-2 infection are rewired by population genetic variants

Diogo Pellegrina, Alexander Bahcheli, Michal Krassowski, and Jüri Reimand

DOI: [10.15252/msb.202110823](https://doi.org/10.15252/msb.202110823)

Corresponding author(s): Jüri Reimand (juri.reimand@utoronto.ca)

Review Timeline:

Submission Date:	19th Nov 21
Editorial Decision:	13th Jan 22
Revision Received:	15th Feb 22
Editorial Decision:	17th Mar 22
Revision Received:	23rd Mar 22
Editorial Decision:	26th Apr 22
Revision Received:	26th Apr 22
Accepted:	27th Apr 22

Editor: Jingyi Hou

Transaction Report:

Thank you for submitting your work to Molecular Systems Biology. We have now heard back from two of the three reviewers who agreed to evaluate your manuscript. Unfortunately, after a series of reminders, we did not obtain a report from Reviewer #3. In the interest of time, we have decided to proceed with the two reports from Reviewers #1 and #2. As you will see from the reports below, the reviewers acknowledge the interest of the study. They raise, however, a series of concerns, which we would ask you to address in a major revision.

Since the reviewers' recommendations are rather clear, there is no need to reiterate all the points listed below. All issues raised by the reviewers need to be satisfactorily addressed. As you may already know, our editorial policy allows in principle a single round of major revision, and it is therefore essential to provide responses to the reviewers' comments that are as complete as possible.

On a more editorial level, we would ask you to address the following issues:

REFeree REPORTS

Reviewer #1:

The manuscript submitted by Pellegrina et al entitled "Human phospho-signaling networks of SARS-CoV-2 infection are rewired by population genetic variants" describes an elegant strategy to aid in deciphering genetic determinants of the severity of symptoms resulting from a SARS-CoV-2 infection. The idea is based on integrating phosphoproteomic data from SARS-CoV-2 infected cell lines with human genetic variation data to infer, which potentially relevant SARS-CoV-2-linked phosphosites might be altered, i.e. disrupted, modified, or gained, by common genetic coding variants. The authors explore these results with respect to the genes primarily targeted by these events, potential pathways involved, potential kinase-substrate switching events, and physical associations between candidate host proteins and viral proteins in a protein-protein interaction (PPI) network.

The manuscript is overall very well written and easy to follow. The figures are clearly labelled and easy to interpret.

Major concerns:

From the results provided, it remains unclear to which extent the reported findings are specific for a SARS-CoV-2 infection. Would one obtain similar target genes and pathways, if the authors used phosphoproteomic data from a different Coronavirus or virus of another family?

As the authors make clear in their discussion, none of the results provide evidence that any of the genetic variants found to overlap with a phosphosite correlate with COVID-19 severity or SARS-CoV-2 infectivity. The authors searched SARS-CoV-2-related GWAS data but as it was to be expected, given the relatively low frequency in the population of identified genetic

variants, no significant signals were obtained. A variety of whole genome and whole exome sequencing data on COVID-19 patients have been published. Would any of these provide a better source of data to look for genotype-phenotype correlations? It remains unclear from the manuscript whether the authors have tried to explore this kind of data to strengthen the relevance of their findings.

In light of this, I would strongly recommend to weaken the title because as it stands it suggests that clear evidence has been revealed for the genetic variants to rewire phospho-signaling networks related to SARS-CoV-2 infections but this is not the case currently.

Minor concerns:

Line 72: I had to read the sentence multiple times to understand what 'differentially phosphorylated' refers to (the phosphosites?). The sentence might be better readable by adding a "that are" in front of the differentially phosphorylated.

Line 106: losses OF kinase binding -> of is missing

Line 110: I am confused by the 87 pSNVs that induced new sequence motifs. So, this means that the phosphosite was not present in the cells that were transfected with SARS-CoV-2 to generate the phosphoproteomic data and thus were not phosphorylated? So, are these 87 pSNVs the result of searching all the genetic variants for cases where they lead to a new phosphosite regardless of SARS-CoV-2 data?

Line 128: lost IN some individuals -> in is missing

Line 199: activation of THE unfolded protein response -> the is missing

Line 202: I suggest rephrasing: ... in core host processes that are important for the virus life cycle.

Line 204: OF tissue-specific gene expression signatures in THE Human Protein Atlas -> of and the are missing

Line 203-220: If I understand this paragraph correctly then tissues were searched in which a significant fraction of the top genes were expressed. If this was the main emphasis of the analysis, then I think that it is somewhat misleading to use the term "tissue-specific gene expression signatures" and "tissue-specific enrichments" to describe the findings. These terms are usually rather used in the context of tissue-specific expression of genes but as the authors state, some of the top genes were highly expressed in most tissues. I suggest rewriting this paragraph to state more clearly what the analysis was about. You could use terms like 'tissue enrichment' rather than tissue-specific enrichment, for example. The same is true for the legend of the corresponding Figure 2E where the authors also refer to tissue-specific expression of genes. Even if the authors refer with that term to the tissues that show an enrichment of expression for the top genes given how many of the tissues in the analysis showed an enrichment, I am not convinced that we can talk about any tissue-specific signal here in any way. Maybe the authors can better clarify?

Figure 2E: What does the 'gene FDR' represent? How strongly a gene was expressed in a given tissue? Please, clarify.

Line 232-233: From the methods section it appears that the random drawing of proteins to estimate the background was not done in a degree-controlled way. This is important since the top genes might have a higher average degree than the rest of the proteins from the network from which other proteins were drawn randomly. Since random drawing of proteins following a degree distribution from the actual set of the top genes is difficult, network scientists rather do degree-controlled network randomizations, i.e. such that edges are rewired but every protein keeps the same number of edges like in the original network. This can be achieved using the `degree_sequence` function from the `igraph` package, for example. Please, clarify or revise the analysis accordingly. If the results will stand, I would recommend mentioning in the text that host-host and kinase-substrate interactions were not more abundant in the real network compared to the random control.

Line 262-263: The sentence starting with "Thus, the predicted motif disruption..." does not make sense grammatically but even when the grammar has been corrected, I'm afraid I don't understand the point that you are trying to make here. Can you please clarify?

Line 270: The title of this section somewhat confused me because I was expecting to read results about motif switching but, unless I misunderstand, until line 297 all results are about motif rewiring cases that are not switching cases. If that is true, I would suggest to change this section title.

Line 302-304: I wonder how significant the finding is that CDK and MAPK motifs change to TBK1 motifs. How likely is this going to happen if a randomly selected phosphosite from the human proteome is randomly mutated?

Reviewer #2:

Pellegrina and co-authors present an analysis of SNVs that potentially impact on protein sites that are differentially phosphorylated upon SARS-COV2 infection (pSNVs). The goal is to identify mechanisms leading to differential responses to SARS-COV2 infections in individuals. Many of the proteins and pathways that are affected are known to be related to innate immunity or even SARS-COV2 infections. Without doubt this is an important and very timely study.

The authors make two discoveries that I find particularly interesting and which seem to be linked: (1) pSNVs resulting in motif rewiring in SARS-COV2-associated phosphosites are enriched compared to a random control (Fig. 1B) and (2) many pSNVs lead to motif switches consistently losing motives in MAPK pathways and re-wiring them as targets of IKK family kinases (Fig. 4C). These two findings are surprising. The first finding is based on comparing the frequency of motif rewiring in phosphosites that are differentially phosphorylated upon SARS-COV2 infection to all (measurable) phosphosites. Hence, the conclusion is that SARS-COV2-affected phosphosites are enriched for sequence variants changing kinase recognition motifs. I can think of only two explanations for this observation: either relaxed selection compared to the background of all phosphosites or positive selection (which seems to be the explanation favored by the authors; Lines 129-130). This notion is supported by Fig. 4C, because the striking consistency of independent pSNVs also suggests some kind of positive selection. My first point of critique is

that this connection between the two observations needs to be strengthened in the Discussion section.

It was a pleasure to read the manuscript. It is very well structured and very clearly written. However, I also have some major concerns:

1. What is the functional relevance of the phosphosites? Some sites clearly seem to have functional relevance (e.g. S473 in TRIM28). However, many of the sequence alterations may have no effect if the phosphosite does not change the activity status of the host protein (see e.g. PMID 31819260). What fraction of the phosphosites is expected to be functionally relevant? This question also links to Figure 5: many of the frequent variants may impact non-functional phosphosites (as also implied by the authors, see Lines 350-351). PMID 31819260 should enable to better separate functional from non-functional sites.
2. The conclusions about evolutionary selection should be strengthened: as I pointed out above, the two findings in Fig. 1B and Fig. 4C strengthen each other. Those variable genomic sites should be analyzed more deeply to strengthen the point about positive versus relaxed selection. In addition to studying the sequence conservation of the phosphosites themselves (previous point) the authors should also investigate the conservation of the kinase recognition sites. Maybe the fact that gnomAD consists of several sub-populations could be exploited (PMID 29459708).
3. I think many of the enrichment analyses are flawed. The problem is that all phosphosites under consideration are differentially phosphorylated upon SARS-COV2 infection. As a consequence, both the host proteins as well as the kinases and regulatory pathways involved are intrinsically enriched for proteins/pathways that are relevant for SARS-COV2 etiology. However, the specific effect of pSNVs for disease etiology may be less clear. Here are some specific examples (there might be more that I have overlooked):
 - a. Lines 116-119: these phosphosites are already pre-selected to be SARS-COV2 affected. However, the control chosen here is the entire human phospho-proteome. Hence, the conclusion is also flawed (lines 124-125): "... in agreement with studies of the human signaling network modulated upon SARS-CoV-2 infection". This is trivial based on how the phosphosites were selected to begin with.
 - b. Lines 206-207: "prioritized genes were often expressed in lung pneumocytes and their squamous epithelial precursors, confirming their relevance to respiratory tissues affected by SARS-CoV-2 infection" Is this really a specific enrichment of phosphosites targeted by pSNVs or just a reflection of differential phosphorylation after SARS-COV2 infection?
 - c. Fig. 3B: "Expected PPI counts were derived from control protein sets drawn randomly from the human phospho-proteome." Same problem again.

Minor points:

4. Figure 1D: these are absolute numbers. How do they relate to the total number of target sites of these kinases. E.g. does CDK1 have an excessively large number of targets in the human proteome and is it hence expected that many of them might also be affected by a pSNV?
5. Line 89: this is a lower bound of the MAF. Did you also define an upper bound? You are frequently writing about 'infrequent variants'. It isn't clear to me if the variants are infrequent by design (i.e. due to pre-selection) or if this is a result of the analysis (i.e. Fig. 5?). Just make the wording explicit.
6. Fig. 2D: The color in the legend (red) does not fit the colors in the circles (orange).
7. Lines 139-140: "the significance scores were adjusted for multiple testing" Corrected for which numbers of tests: number of phosphosites per protein or number of SNVs per protein? Or both?
8. Figure 4B: panel for CAMK2A is missing (see figure caption).

Reviewer #1:

The manuscript submitted by Pellegrina et al entitled "Human phospho-signaling networks of SARS-CoV-2 infection are rewired by population genetic variants" describes an elegant strategy to aid in deciphering genetic determinants of the severity of symptoms resulting from a SARS-CoV-2 infection. The idea is based on integrating phosphoproteomic data from SARS-CoV-2 infected cell lines with human genetic variation data to infer, which potentially relevant SARS-CoV-2-linked phosphosites might be altered, i.e. disrupted, modified, or gained, by common genetic coding variants. The authors explore these results with respect to the genes primarily targeted by these events, potential pathways involved, potential kinase-substrate switching events, and physical associations between candidate host proteins and viral proteins in a protein-protein interaction (PPI) network.

The manuscript is overall very well written and easy to follow. The figures are clearly labelled and easy to interpret.

Thank you for the summary and positive comments!

Major concerns:

1. From the results provided, it remains unclear to which extent the reported findings are specific for a SARS-CoV-2 infection. Would one obtain similar target genes and pathways, if the authors used phosphoproteomic data from a different Coronavirus or virus of another family?

Thank for a great point. We now compared SARS-CoV-2 associated phosphosites with two additional sets of phosphosites detected in human immunodeficiency virus (HIV) infection and in herpesvirus (HSV-1) infection. In both cases, the other phosphosites were mostly distinct from SARS-CoV-2 associated phosphosites. Approximately 17% of SARS-CoV-2 associated sites were shared, and ~15% of sites with pSNVs were shared. Therefore, our findings are relatively specific to SARS-CoV-2 infections. This analysis has been added to **Figure 1F-G**.

2. As the authors make clear in their discussion, none of the results provide evidence that any of the genetic variants found to overlap with a phosphosite correlate with COVID-19 severity or SARS-CoV-2 infectivity. The authors searched SARS-CoV-2-related GWAS data but as it was to be expected, given the relatively low frequency in the population of identified genetic variants,

Review response of MSB-2021-10823:
 Human phospho-signaling networks of SARS-CoV-2 infection are rewired by population genetic variants

no significant signals were obtained. A variety of whole genome and whole exome sequencing data on COVID-19 patients have been published. Would any of these provide a better source of data to look for genotype-phenotype correlations? It remains unclear from the manuscript whether the authors have tried to explore this kind of data to strengthen the relevance of their findings.

Thank you for the comment. This helped improve our manuscript and led to new interesting findings. We studied Regeneron exome sequencing data that evaluated COVID-19 outcomes across several cohorts (UK Biobank, AncestryDNA, Geisinger, Penn) (PMID: 34115965).

We found 10 pSNVs that associated with severe disease and hospitalization using a less-stringent restricted false discovery rate filter that only focused on the SNVs that we identified in our analysis (restricted FDR < 0.1).

The results included one pSNV in STARD13 with a potentially protective effect. This pSNV was sufficiently frequent in the population that it was also detectable in the GWAS study of the COVID-19 Host Genetics Initiative, where it also showed nominal statistical significance and a potentially protective effect.

These results provide proof-of-concept to our analysis, however the relatively weak statistical effects warrant caution in interpretation. We have added these results to **Figure 5A-C**.

Review response of MSB-2021-10823:

Human phospho-signaling networks of SARS-CoV-2 infection are rewired by population genetic variants

In light of this, I would strongly recommend to weaken the title because as it stands it suggests that clear evidence has been revealed for the genetic variants to rewire phospho-signaling networks related to SARS-CoV-2 infections but this is not the case currently.

We have now added evidence that several pSNVs associate with COVID-19 risk. We politely suggest that we have more support to maintain the originally proposed title. However, we are open to changing the title if necessary.

Minor concerns:

1. Line 72: I had to read the sentence multiple times to understand what 'differentially phosphorylated' refers to (the phosphosites?). The sentence might be better readable by adding a "that are" in front of the differentially phosphorylated.

OK, thanks for pointing it out. Adding "that are" does make it easier to read.

2. Line 106: losses OF kinase binding -> of is missing

Thanks – we fixed this one too.

3. Line 110: I am confused by the 87 pSNVs that induced new sequence motifs. So, this means that the phosphosite was not present in the cells that were transfected with SARS-CoV-2 to generate the phosphoproteomic data and thus were not phosphorylated? So, are these 87 pSNVs the result of searching all the genetic variants for cases where they lead to a new phosphosite regardless of SARS-CoV-2 data?

We added "potential disruption of phosphorylation" instead of "loss of phosphorylation", and "gain of phosphorylation by a specific kinase" instead of "gain of phosphorylation".

To clarify, all phosphosites we analyze are differentially phosphorylated upon SARS-CoV-2 infection. Many of these sites do not have a consensus sequence motif in the reference protein sequence, however a motif is created when a genetic variant (amino acid substitution) is considered.

4. Line 128: lost IN some individuals -> in is missing
5. Line 199: activation of THE unfolded protein response -> the is missing
6. Line 202: I suggest rephrasing: ... in core host processes that are important for the virus life cycle.
7. Line 204: OF tissue-specific gene expression signatures in THE Human Protein Atlas -> of and the are missing

Thank you – those four sentences have been fixed.

8. Line 203-220: If I understand this paragraph correctly then tissues were searched in which a significant fraction of the top genes were expressed. If this was the main

Review response of MSB-2021-10823:

Human phospho-signaling networks of SARS-CoV-2 infection are rewired by population genetic variants

emphasis of the analysis, then I think that it is somewhat misleading to use the term "tissue-specific gene expression signatures" and "tissue-specific enrichments" to describe the findings. These terms are usually rather used in the context of tissue-specific expression of genes but as the authors state, some of the top genes were highly expressed in most tissues. I suggest rewriting this paragraph to state more clearly what the analysis was about. You could use terms like 'tissue enrichment' rather than tissue-specific enrichment, for example. The same is true for the legend of the corresponding Figure 2E where the authors also refer to tissue-specific expression of genes. Even if the authors refer with that term to the tissues that show an enrichment of expression for the top genes given how many of the tissues in the analysis showed an enrichment, I am not convinced that we can talk about any tissue-specific signal here in any way. Maybe the authors can better clarify?

Thank you for pointing out this lack of clarity. We agree that the word *tissue-specific* was not appropriate in this context since we did not mean to indicate genes that are exclusively expressed in some tissue and not expressed in any other tissues. We have rephrased it in the legend of **Figure 2E** and the Results section as suggested. We now refer to the analysis as *gene expression signatures of various tissues*. Hopefully this makes more sense to readers.

9. Figure 2E: What does the 'gene FDR' represent? How strongly a gene was expressed in a given tissue? Please, clarify.

Sorry for this lack of clarity. The FDR values in **Figure 2E** reflect the pSNV impact scores that are also shown in panel A. We have clarified this in Figure 2 and the legend.

10. Line 232-233: From the methods section it appears that the random drawing of proteins to estimate the background was not done in a degree-controlled way. This is important since the top genes might have a higher average degree than the rest of the proteins from the network from which other proteins were drawn randomly. Since random drawing of proteins following a degree distribution from the actual set of the top genes is difficult, network scientists rather do degree-controlled network randomizations, i.e. such that edges are rewired but every protein keeps the same number of edges like in the original network. This can be achieved using the `degree_sequence` function from the `igraph` package, for example. Please, clarify or revise the analysis accordingly. 'If the results will stand, I would recommend mentioning in the text that host-host and kinase-substrate interactions were not more abundant in the real network compared to the random control.

Thank for this excellent comment. We have now revised this analysis as suggested. We also addressed the comment by the other reviewer who recommended that we use the phosphoproteome of SARS-CoV-2 infection to construct control networks, instead of all human phosphoproteins (**Reviewer 2, Comment 3-C**; see below).

We simulated control networks based on the node degree distribution of the real protein-protein interaction network, using the `iGraph` package in R and the `sample_degseq()` function. The network only included all human proteins with any SARS-CoV-2 infection associated phosphorylation sites, as well as the SARS-CoV-2 proteins and kinases with site-specific interactions and pSNVs. Expected counts of interactions were derived from 10,000 simulated networks.

Review response of MSB-2021-10823:

Human phospho-signaling networks of SARS-CoV-2 infection are rewired by population genetic variants

We replicated the finding of enriched human-virus PPIs shown in the earlier study and extended the results by showing that the pSNV-prioritized top proteins themselves tend to interact less frequently than expected from the control networks.

This analysis is now shown in the updated manuscript (**Figure 3B**). We updated the Results section, also noting the fewer PPIs among human top proteins and the sub-significant over-representation among kinase-substrate interactions.

11. Line 262-263: The sentence starting with "Thus, the predicted motif disruption..." does not make sense grammatically but even when the grammar has been corrected, I'm afraid I don't understand the point that you are trying to make here. Can you please clarify?

Thank you. This sentence is indeed unclear. We have updated this section of text and hopefully improved the text.

12. Line 270: The title of this section somewhat confused me because I was expecting to read results about motif switching but, unless I misunderstand, until line 297 all results are about motif rewiring cases that are not switching cases. If that is true, I would suggest to change this section title.

Thanks for a great comment that helped improve the structure of the manuscript. We have split this section into two smaller sections, first covering the overall motif-rewiring networks and then describing the motif-switching aspect. Separate section titles were added.

13. Line 302-304: I wonder how significant the finding is that CDK and MAPK motifs change to TBK1 motifs. How likely is this going to happen if a randomly selected phosphosite from the human proteome is randomly mutated?

Thanks for this great comment. To address the comment and derive expected counts of these motif-switching mutations, we studied random sets of PTM sites sampled from the overall human phosphoproteome. This is an extension of the analysis shown in Figure 1B.

Review response of MSB-2021-10823:

Human phospho-signaling networks of SARS-CoV-2 infection are rewired by population genetic variants

We confirmed that motif-switching mutations in SARS-CoV-2 phosphosites occur more frequently than expected. Specifically, the observed number of motif-switching pSNVs overall, the numbers of pSNVs replacing CDK/MAPK motifs with TBK1 motifs, and the numbers of pSNVs causing gains of TBK1 motifs (incl the motif-switching ones) are all significantly higher than expected from equally sized sets of PTM sites sampled from the phosphoproteome. This is now reported in the manuscript (**Figure 4D**).

Review response of MSB-2021-10823:

Human phospho-signaling networks of SARS-CoV-2 infection are rewired by population genetic variants

Reviewer #2:

Pellegrina and co-authors present an analysis of SNVs that potentially impact on protein sites that are differentially phosphorylated upon SARS-COV2 infection (pSNVs). The goal is to identify mechanisms leading to differential responses to SARS-COV2 infections in individuals. Many of the proteins and pathways that are affected are known to be related to innate immunity or even SARS-COV2 infections. Without doubt this is an important and very timely study.

Thank you for the positive comments and nice summary.

The authors make two discoveries that I find particularly interesting and which seem to be linked: (1) pSNVs resulting in motif rewiring in SARS-COV2-associated phosphosites are enriched compared to a random control (Fig. 1B) and (2) many pSNVs lead to motif switches consistently losing motives in MAPK pathways and re-wiring them as targets of IKK family kinases (Fig. 4C). These two findings are surprising. The first finding is based on comparing the frequency of motif rewiring in phosphosites that are differentially phosphorylated upon SARS-COV2 infection to all (measurable) phosphosites. Hence, the conclusion is that SARS-COV2-affected phosphosites are enriched for sequence variants changing kinase recognition motifs. I can think of only two explanations for this observation: either relaxed selection compared to the background of all phosphosites or positive selection (which seems to be the explanation favored by the authors; Lines 129-130). This notion is supported by Fig. 4C, because the striking consistency of independent pSNVs also suggests some kind of positive selection. My first point of critique is that this connection between the two observations needs to be strengthened in the Discussion section. It was a pleasure to read the manuscript. It is very well structured and very clearly written. However, I also have some major concerns:

The comment on positive vs. related selection is especially interesting. We have added a paragraph in the Discussion section of our manuscript. We also added the recommended analysis of positively selected regions (see below; **Comment 2**).

1. What is the functional relevance of the phosphosites? Some sites clearly seem to have functional relevance (e.g. S473 in TRIM28). However, many of the sequence alterations may have no effect if the phosphosite does not change the activity status of the host protein (see e.g. PMID 31819260). What fraction of the phosphosites is expected to be functionally relevant? This question also links to Figure 5: many of the frequent variants may impact non-functional phosphosites (as also implied by the authors, see Lines 350-351). PMID 31819260 should enable to better separate functional from non-functional sites.

Thanks for the comment and recommendation. We studied the functional relevance scores of SARS-CoV-2 associated phosphosites with pSNVs relative to other phosphosites in the overall human phosphoproteome. SARS-CoV-2 associated phosphosites with pSNVs showed significantly higher functional relevance scores compared to non-SARS-CoV-2 associated phosphosites in the human proteome (Wilcoxon log-rank test, $P = 10^{-110}$). Therefore, many of the pSNVs affect potentially functional phosphosites that may alter protein activity, lending confidence to our findings.

- phosphosites at all pSNVs; median score = 0.44 (plot below; left)
- control phosphosites, human proteome; median score = 0.32 (plot below; right)

Review response of MSB-2021-10823:

Human phospho-signaling networks of SARS-CoV-2 infection are rewired by population genetic variants

In response to the second part of the question, there was no significant statistical difference between high allele frequency pSNVs ($AF_{popmax} > 1\%$) and the pSNVs with lower allele frequency (Wilcoxon $P = 0.11$). The scores of the high-AF variants were indeed somewhat smaller compared to the low-AF variants, as predicted by the Reviewer.

- sites at pSNVs with $AF_{popmax} > 1\%$; median func_score = 0.41
- sites at pSNVs with $AF_{popmax} < 1\%$; median func_score = 0.45

2. The conclusions about evolutionary selection should be strengthened: as I pointed out above, the two findings in Fig. 1B and Fig. 4C strengthen each other. Those variable genomic sites should be analyzed more deeply to strengthen the point about positive versus relaxed selection. In addition to studying the sequence conservation of the phosphosites themselves (previous point) the authors should also investigate the conservation of the kinase recognition sites. Maybe the fact that gnomAD consists of several sub-populations could be exploited (PMID 29459708).

Great comment. To address this point we studied positive selection of genes with pSNVs using the dataset referenced above. Indeed, >20% of these genes showed evidence of positive selection in some human populations, including many of the top genes with pSNVs we prioritized. We added this result to **Figure 5E** and extended the Discussion section.

3. I think many of the enrichment analyses are flawed. The problem is that all phosphosites under consideration are differentially phosphorylated upon SARS-COV2 infection. As a

Review response of MSB-2021-10823:

Human phospho-signaling networks of SARS-CoV-2 infection are rewired by population genetic variants

consequence, both the host proteins as well as the kinases and regulatory pathways involved are intrinsically enriched for proteins/pathways that are relevant for SARS-COV2 etiology. However, the specific effect of pSNVs for disease etiology may be less clear.

Thank you, this is an interesting and relevant comment. If we understand the comment correctly, it points out that the statistical background set (*i.e.*, the control) of all human phosphoproteins is suboptimal for some of our analyses and another protein set related to SARS-CoV-2 infection should be used instead.

Overall, we studied a relatively few human phosphoproteins (949) that included at least one differentially phosphorylated site according to the SARS-CoV-2 proteomics study (Bouhaddou,2020). Most of those proteins also included at least one pSNV (693 proteins, 73%). Therefore, an alternative control set of proteins would additionally include SARS-CoV-2-infection phosphorylated but not mutated proteins (949 - 693 = 256). This additional set would be rather small and underpowered for most enrichment analyses and not distinctive from the proportionally larger set of phosphorylated and mutated proteins.

The three scenarios described below are distinct and we have addressed these separately.

Here are some specific examples (there might be more that I have overlooked):

a. Lines 116-119: these phosphosites are already pre-selected to be SARS-COV2 affected. However, the control chosen here is the entire human phospho-proteome. Hence, the conclusion is also flawed (lines 124-125): "... in agreement with studies of the human signaling network modulated upon SARS-CoV-2 infection". This is trivial based on how the phosphosites were selected to begin with.

This is the overall analysis of all pSNVs affecting SARS-CoV-2 associated phosphosites (**Figure 1**). We propose that in this case, the comparison to the overall human phosphoproteome is appropriate. Here, we point out an abundance of motif-rewiring mutations in the signalling networks responding to the virus infection. This analysis shows that the SARS-CoV-2 infection affects a specific component of the overall human phospho-signalling network in which genetic variants with motif-altering impact are particularly frequent.

To control for the number of phosphosites and estimate the variation in the background expectation, we have sampled equal numbers of phosphosites from the human phosphoproteome and annotated these with SNVs in gnomAD, using the same procedure as for SARS-CoV-2 associated phosphosites.

With regards to the second part of the comment regarding the conclusion of the paragraph (lines 124-125), our text was somewhat convoluted, and we have revised it. We meant to say that many known binding sites of kinases such as CDKs, casein kinases, CHEKs are detected at pSNVs in our analysis. These known sites validate our mutation analysis since these kinases are prominently detected in the study by Bouhaddou et al (hence the reference to this study). Our analysis of mutations complements the phosphoproteomics study, however the recovery of these known kinases in our analysis offers additional confidence in our findings. Also, later in the manuscript we show that the same families of kinases are also associated with pSNVs due to altered sequence motifs (**Figure 4A-B**).

We now show that the known binding sites of these kinases are enriched compared to the phosphoproteome, in response to **Comment 4** (see below). This is shown in **Figure 1D**.

Review response of MSB-2021-10823:

Human phospho-signaling networks of SARS-CoV-2 infection are rewired by population genetic variants

b. Lines 206-207: "prioritized genes were often expressed in lung pneumocytes and their squamous epithelial precursors, confirming their relevance to respiratory tissues affected by SARS-CoV-2 infection" Is this really a specific enrichment of phosphosites targeted by pSNVs or just a reflection of differential phosphorylation after SARS-COV2 infection?

For clarification, this analysis considers the prioritized set of 77 genes ranked by functional scoring of pSNVs (FDR < 0.1). Similar pathway enrichment analysis techniques are considered to analyze biological pathways and processes (**Figure 2D**) and highly expressed genes in various tissues (**Figure 2E**). The statistical background set includes ~14,000 human phosphoproteins.

We agree with the reviewer that in this case, an alternative analysis could consider the top-ranked proteins/genes relative to all 949 phosphoproteins with differential phosphosites in SARS-CoV-2 infection. However, we politely suggest that the current analysis should be used in our manuscript for the reasons described below.

We tested this alternative background set of 949 phosphoproteins in an enrichment analysis and indeed found that some of the major processes, pathways and tissues of the original analysis were recovered at the top of the list with nominally significant P-values ($P < 0.01$; asterisks show pathways also detected in the main analysis; **Figure** below). Thus, the major functional themes of Ras/Rho signalling and alternative splicing remain to be associated with pSNVs even if this more stringent approach is used.

However, the narrower background set was less powered statistically, due to the strong effect of multiple testing correction (thousands of gene sets corresponding to pathways, processes, and tissue-expressed genes were tested) and the smaller effect sizes of more specific gene sets that remained undetectable in this constrained analysis (such as the GLUT4 translocation pathway we report in our study).

Review response of MSB-2021-10823:

Human phospho-signaling networks of SARS-CoV-2 infection are rewired by population genetic variants

Our global background set of 14,000 phosphoproteins is also justified in the experimental context. The phosphosites were originally detected in SARS-CoV-2 infection of Vero6 cells of the kidney of green monkey. The pSNVs and proteins we identify, and thus the SARS-CoV-2-associated phosphoproteins in general, are significantly enriched in core pathways, processes, protein complexes and diverse human tissues relevant for SARS-CoV-2 infection and COVID-19, even though the model system used to generate the data is limited to one type of non-human tissue. This supports our findings of pSNVs as well as the earlier phosphoproteomics experiments enabling our analysis.

To partially address this comment, we updated our pathway and tissue-expression analysis using more stringent filtering threshold ($FDR < 0.05$). This more stringent analysis led to the removal of some less-supported pathways and tissues. The updated analysis is now shown in **Figure 2D-E**.

c. Fig. 3B: "Expected PPI counts were derived from control protein sets drawn randomly from the human phospho-proteome." Same problem again.

For clarification, this part is about the protein-protein interaction networks involving top genes with pSNVs, proteins encoded by SARS-CoV-2, and kinases phosphorylating specific phosphosites with pSNVs.

As recommended in the comment, we have updated the background set of proteins to 949 SARS-CoV-2 associated phosphoproteins and limited the background network of PPIs. We also further developed this analysis to address another comment (**Reviewer 1, minor Comment 10**) using degree-informed network permutations instead of drawing control proteins randomly.

The analysis confirms that human-virus PPIs are enriched among top proteins with pSNVs. In addition, we find that human-human PPIs occur less frequently among the top proteins than

Review response of MSB-2021-10823:

Human phospho-signaling networks of SARS-CoV-2 infection are rewired by population genetic variants

expected. Also, site-specific kinase-substrate interactions with pSNVs are somewhat over-represented at a sub-significant level. This analysis is now shown in **Figure 3B**.

Minor points:

4. Figure 1D: these are absolute numbers. How do they relate to the total number of target sites of these kinases. E.g. does CDK1 have an excessively large number of targets in the human proteome and is it hence expected that many of them might also be affected by a pSNV?

We have updated this analysis and evaluated the expected frequency of these known kinase binding sites by sampling equivalent sets of phosphosites from the human phosphoproteome, similarly to the overall pSNV impact analysis shown earlier in this figure. Nearly all the top kinases occur more often near pSNVs than expected. The updated analysis is shown in **Figure 1D** and copied below.

5. Line 89: this is a lower bound of the MAF. Did you also define an upper bound? You are frequently writing about 'infrequent variants'. It isn't clear to me if the variants are infrequent by design (i.e. due to pre-selection) or if this is a result of the analysis (i.e. Fig. 5?). Just make the wording explicit.

Thank you for pointing it out, we have reviewed the wording in the manuscript. We only defined a lower bound (maximum population frequency of 10^{-4}) to avoid the very rare variants as their number was considerably larger. To identify frequent variants, we did not define an upper bound

Review response of MSB-2021-10823:

Human phospho-signaling networks of SARS-CoV-2 infection are rewired by population genetic variants

for allele frequency. Our analysis shows that most pSNVs are closer to the lower bound than the upper bound of allele frequency. This is quite expected as most variants in human genomes are relatively infrequent.

We emphasize low variant frequency throughout the manuscript to encourage a more cautious interpretation of our study in the context of immediate disease implications.

6. Fig. 2D: The color in the legend (red) does not fit the colors in the circles (orange).

Thanks for pointing it out – we have fixed the color legend in the updated enrichment map.

7. Lines 139-140: "the significance scores were adjusted for multiple testing" Corrected for which numbers of tests: number of phosphosites per protein or number of SNVs per protein? Or both?

The FDR correction was based on genes after gene-based significance was computed across all pSNVs per gene. pSNVs were quantified as posterior probabilities and FDR corrections of those values are not appropriate. For increased stringency, FDR was conducted across all phosphoproteins assuming $P=1$ for genes lacking SARS-CoV-2 phosphosites. We reviewed the text in Results and Methods for clarity.

8. Figure 4B: panel for CAMK2A is missing (see figure caption).

Thank you for a great catch. We apologize for missing this outdated section of the figure caption. Since our figures have become quite busy, we decided to exclude this panel and deleted the part of the caption.

Thank you for sending us your revised manuscript. We have now heard back from the two reviewers who agreed to evaluate your study. As you will see below, the reviewers acknowledge the effort that has been made to revise the manuscript. However, Reviewer #2 raises significant concerns about a number of issues that were already brought up during the first round of review, such as the enrichment analysis and statistical tests, and thought several main conclusions of the study could not be supported by the presented analyses. Furthermore, this reviewer rated technical quality, adequacy of method analysis and validation as "Low" and explicitly indicated that they do not recommend publication in Molecular Systems Biology.

Based on these comments, we think the raised concerns are substantial and remain insufficiently addressed. Under these circumstances and given that our editorial policy is in principle to allow a single round of major revision, I see no other choice than to return the manuscript with the message that we cannot offer to publish it.

I am very sorry that I cannot bring better news on this occasion. I hope that the points raised in the reports will prove helpful to you and that you will not be discouraged from submitting future work to Molecular Systems Biology.

Reviewer #1:

All comments have been addressed to my satisfaction.

Reviewer #2:

In response to the reviewer comments Pellegrina and co-authors have re-worked the manuscript and added some new analyses, including new data. While many of the comments of the two reviewers were sufficiently addressed, I still have some remaining major concerns that mainly regard the statistical analysis of the data. In particular, I am concerned that some of the conclusions drawn are not supported by the statistical analyses. The reasoning of the authors in their response letter is worrisome to me. Dismissing a statistical test because it does not deliver the desired result is not scientifically sound. I don't know if that was the intention of the authors (I hope not), but the wording chosen severely questions the quality of the analysis.

Major concerns

1. Enrichment analysis is done with a phosphoproteome wide background that is not suitable to answer the questions asked by the authors. I do agree with the authors that in cases where SNVs inside proteins with differentially phosphorylated sites are compared to the background of SNVs in phosphoproteins such testing is okay. However, a more specific background is necessary when trying to investigate what kind of phosphoproteins have pSNVs (Section starting at line 212). By definition the phosphosites in these proteins have to be responsive to the infection. The enrichment of immune functions among proteins with pSNVs is likely already present in the selected phosphosites and therefore the analysis does not demonstrate if it is due to the presence of SNVs. When trying to investigate what kinds of proteins have pSNVs one should compare them to SARS-COV-2 responsive phosphoproteins without pSNVs.

The reasoning of the authors on this aspect in their point-by-point response is worrisome:

"Overall, we studied a relatively few human phosphoproteins (949) that included at least one differentially phosphorylated site according to the SARS-CoV-2 proteomics study (Bouhaddou,2020). Most of those proteins also included at least one pSNV (693

proteins, 73%). Therefore, an alternative control set of proteins would additionally include SARS-CoV-2-infection phosphorylated but not mutated proteins (949 - 693 = 256). This additional set would be rather small and underpowered for most enrichment analyses and not distinctive from the proportionally larger set of phosphorylated and mutated proteins."

Further down they go on:

"[...] We tested this alternative background set of 949 phosphoproteins in an enrichment analysis and indeed found that some of the major processes, pathways and tissues of the original analysis were recovered at the top of the list with nominally significant P-values ($P < 0.01$; asterisks show pathways also detected in the main analysis; Figure below). Thus, the major functional themes of Ras/Rho signalling and alternative splicing remain to be associated with pSNVs even if this more stringent approach is used.

However, the narrower background set was less powered statistically, due to the strong effect of multiple testing correction (thousands of gene sets corresponding to pathways, processes, and tissue-expressed genes were tested) and the smaller effect sizes of more specific gene sets that remained undetectable in this constrained analysis (such as the GLUT4 translocation pathway we report in our study)."

If a particular statistical test is underpowered one cannot simply switch the test scheme. The test has to be appropriate to address the scientific question being asked. (Note that simply using a more stringent FDR cutoff does not solve the problem that the statistical test is inappropriate in the first place.) Here, the analysis focuses on the specific features of phosphosites that are differentially phosphorylated upon SARS-COV-2 infection and hosting a pSNV. Dismissing a statistical test because it does not deliver the desired result is not scientifically sound.

2. The statistical testing done in this work is at times either not supporting the conclusions or not described sufficiently. It is unclear to me how the permutations for motif switching were done exactly. The methods are not precise enough here. (The following statement was added: "The expected numbers of other functional annotations we assessed similarly: all motif switching pSNVs, CDK/MAPK to TBK1 motif switches, and pSNVs in kinase binding sites.") Is a given CDK/MAPK motif more likely to switch to TBK if the protein is responsive to a SARS-COV-2 infection or are just more proteins with CDK/MAPK motifs responsive to a SARS-COV-2 infection on the phospho level? A simple Fisher's test might be sufficient instead of permutations. In my previous review I was quite excited about the observation that MAPK/ERK1 motifs were consistently switching to IKK and PIKK motifs. However, one has to carefully assess if this could happen by chance. Obviously, all of the motifs are very similar, since switching a single site suffices to switch the motif. If in the relevant protein set there are many more MAPK/ERK1 motifs compared to IKK motifs to begin with, it is more likely (simply by chance) that they get 'hit' by a polymorphism, which might switch that motif towards IKK. The description in the Methods section is very slim and does not clarify if this potential bias has been considered and corrected for.

3. The authors tend to use the word "affect" loosely. It should only be used to report known facts. A phosphosite being predicted to switch a motif, doesn't per se justify the usage of "affect" (e.g. line 286) when indicating that a pSNVs is located at a phosphosite. In my interpretation the use of the word "affect" implies that the SNV was shown to change the phosphorylation of this site. In this work effects on phosphorylation are predicted and not validated, therefore I would advise against the use of this phrase in this context. In the case of phosphosites where the phosphorylated residue itself is mutated it would be acceptable to use the term.

Minor points

4. Regarding point 1 from Reviewer 1: Here the reviewer was asking to compare against phospho data from other viruses. The authors' reply was as follows:

"Thank for a great point. We now compared SARS-CoV-2 associated phosphosites with two additional sets of phosphosites detected in human immunodeficiency virus (HIV) infection and in herpesvirus (HSV-1) infection. In both cases, the other phosphosites were mostly distinct from SARS-CoV-2 associated phosphosites. Approximately 17% of SARS-CoV-2 associated sites were shared, and ~15% of sites with pSNVs were shared. Therefore, our findings are relatively specific to SARS-CoV-2 infections. This analysis has been added to Figure 1F-G."

I appreciate the inclusion of additional data in the analysis. However, the first observation (only 17% of the phospho-sites overlapped with the other data) is a feature of the originally published phospho-proteomics data, i.e. unrelated to the question of genomic variability. The second result (15% with pSNVs were shared; i.e. 88% out of the 17% from above) is not surprising in light of the fact that 73% of the phospho-sites have a pSNV. 88% is still more than 73%, but it is unclear if this apparent enrichment could happen by chance.

5. In Figure 1a the diamonds symbols are only present in the legend.

6. In Figure 1d all shown kinases are overrepresented. How can that be? Does the set of responsive phosphoproteins have more high quality motifs in general? If yes that has to be controlled.

7. A sentence in line 231 seems incoherent: "Genes with frequent pSNVs genes"

8. A word is missing in line 281: "TBC1D4 (i.e., AS160) is a GTPase-activating protein"

9. I fail to grasp the argument in line 280 and following. Are the authors arguing that the pSNV increases the risk of severe COVID through an increase in the risk of diabetes type II?

10. In line 304 the authors state that 217 pSNVs correspond to 3360 predictions of motif rewiring. This would correspond to around 15 motif rewirings per site. This could be written more clearly.
11. In line 359 the authors state that the gene ARFGEF2 causes several disease phenotypes. Do they mean that a gain, full loss or partial loss of function of ARFGEF2 causes this phenotype? How does the SNV relate to this?
12. How can the pSNV P257L alter a phosphosite (a serine) also at position S257 (lines 391-392)

** As a service to authors, EMBO Press offers the possibility to directly transfer declined manuscripts to another EMBO Press title or to the open access journal Life Science Alliance launched in partnership between EMBO Press, Rockefeller University Press and Cold Spring Harbor Laboratory Press. The full manuscript and if applicable, reviewers' reports, are automatically sent to the receiving journal to allow for fast handling and a prompt decision on your manuscript. For more details of this service, and to transfer your manuscript please click on Link Not Available. **

Reviewer #1:

All comments have been addressed to my satisfaction.

We would like to thank the Reviewer for approving our manuscript.

Reviewer #2:

In response to the reviewer comments Pellegrina and co-authors have re-worked the manuscript and added some new analyses, including new data. While many of the comments of the two reviewers were sufficiently addressed, I still have some remaining major concerns that mainly regard the statistical analysis of the data. In particular, I am concerned that some of the conclusions drawn are not supported by the statistical analyses. The reasoning of the authors in their response letter is worrisome to me. Dismissing a statistical test because it does not deliver the desired result is not scientifically sound. I don't know if that was the intention of the authors (I hope not), but the wording chosen severely questions the quality of the analysis.

Thanks for the additional review and constructive comments. We have revised the manuscript and added some analyses. We apologise ahead for the long discussion and are grateful for the time and effort of the Reviewer. This has certainly helped improve our manuscript. We have underlined certain parts of the Reviewer's comments that we refer to in our response.

Major concerns

1. Enrichment analysis is done with a phosphoproteome wide background that is not suitable to answer the questions asked by the authors. I do agree with the authors that in cases where SNVs inside proteins with differentially phosphorylated sites are compared to the background of SNVs in phosphoproteins such testing is okay.

However, a more specific background is necessary when trying to investigate what kind of phosphoproteins have pSNVs (Section starting at line 212). By definition the phosphosites in these proteins have to be responsive to the infection. The enrichment of immune functions among proteins with pSNVs is likely already present in the selected phosphosites and therefore the analysis does not demonstrate if it is due to the presence of SNVs. When trying to investigate what kinds of proteins have pSNVs one should compare them to SARS-COV-2 responsive phosphoproteins without pSNVs. The reasoning of the authors on this aspect in their point-by-point response is worrisome:

"Overall, we studied a relatively few human phosphoproteins (949) that included at least one differentially phosphorylated site according to the SARS-CoV-2 proteomics study (Bouhaddou,2020). Most of those proteins also included at least one pSNV (693 proteins, 73%). Therefore, an alternative control set of proteins would additionally

RE: Manuscript MSB-2021-10823R, Human phospho-signaling networks of SARS-CoV-2 infection are rewired by population genetic variants

include SARS-CoV-2-infection phosphorylated but not mutated proteins (949 - 693 = 256). This additional set would be rather small and underpowered for most enrichment analyses and not distinctive from the proportionally larger set of phosphorylated and mutated proteins."

Further down they go on:

"[...] We tested this alternative background set of 949 phosphoproteins in an enrichment analysis and indeed found that some of the major processes, pathways and tissues of the original analysis were recovered at the top of the list with nominally significant P-values ($P < 0.01$; asterisks show pathways also detected in the main analysis; Figure below). Thus, the major functional themes of Ras/Rho signalling and alternative splicing remain to be associated with pSNVs even if this more stringent approach is used. However, the narrower background set was less powered statistically, due to the strong effect of multiple testing correction (thousands of gene sets corresponding to pathways, processes, and tissue-expressed genes were tested) and the smaller effect sizes of more specific gene sets that remained undetectable in this constrained analysis (such as the GLUT4 translocation pathway we report in our study)."

If a particular statistical test is underpowered one cannot simply switch the test scheme. The test has to be appropriate to address the scientific question being asked. (Note that simply using a more stringent FDR cutoff does not solve the problem that the statistical test is inappropriate in the first place.) Here, the analysis focuses on the specific features of phosphosites that are differentially phosphorylated upon SARS-COV-2 infection and hosting a pSNV. Dismissing a statistical test because it does not deliver the desired result is not scientifically sound.

Thank you for the comment and discussion. We have now revised these analyses using the background set of phosphoproteins from the (Bouhaddou, 2020) study, as recommended above and in revision 1. This part concerns the pathway enrichment analysis (**Figure 2D**) and the enrichment analysis of gene expression signatures in human tissues (**Figure 2E**).

In the pathway analysis, the functional themes we reported previously are still detected in the new analysis.

- Spliceosome complex representing the RNA splicing theme
- Small GTPase, Ras and Rho signalling
- Golgi to endoplasmic reticulum transport
- CEN complex
- Translocation of GLUT4 to the plasma membrane
- Microtubule based transport

Note that larger RNA-processing gene sets were removed due to the size-based gene set filters. On the other hand, several gene sets related to microtubule-based transport were recovered again (we found those in our original manuscript).

Thus, our major findings are relatively robust to this more stringent definition of background genes/proteins, and the recurrence of these major functional themes in different versions and gene universe definitions of the enrichment analysis strengthens these results. See the updated **Figure 2D** below.

Two major changes were made in the underlying analysis pipeline:

- Selecting smaller, more specific gene sets for statistical testing to tackle the major impact of stringent multiple-testing correction due to redundant gene sets.
- Updating the FDR threshold to the more lenient $FDR < 0.2$ (previously $FDR < 0.1$ in original analysis; $FDR < 0.05$ in revision 1).

Selecting smaller, more specific functional gene sets was the key to this solution. Please see a detailed description of the problem and the solution below.

Since we are testing a limited gene list including 693 genes with pSNVs and altogether 949 genes/proteins in the gene universe, the original analysis was underpowered. It was greatly influenced by the large number of functional gene sets we that tested initially (>11,000 biological processes, gene functions, molecular pathways, protein complexes). As many of the pathways we analysed were of similar size or larger than our gene background, these were not specific to our smaller dataset and their presence in the analysis needlessly increased the effect of multiple testing correction.

Gene Ontology is built in a redundant way so that smaller gene sets (i.e., specific biological processes) are entirely included in larger gene sets (i.e., more general biological processes) by design. This leads to many statistical tests and a greatly reduced statistical power initially due to multiple testing correction. In GO, most gene sets are very small and specific (~1-5 genes) while some are very large and general and cover a large fraction of genes (up to 80-100%). Neither very small nor very large

gene sets are useful for enrichment analysis in most cases. This is discussed in our recent review - <https://www.nature.com/articles/s41596-018-0103-9>.

To overcome this challenge, we focused our analysis on only the gene sets that were relatively specific and sufficiently represented in the SARS-CoV-2 associated phosphoproteins. For GO, Reactome, and CORUM, we selected gene sets that had between 10 and 50 genes among all the SARS-CoV-2 associated phosphoproteins. This reduced the number of tests from ~11,000 to ~1,300, greatly reducing the effect of multiple testing correction and increasing our statistical power.

The analysis of gene expression signatures of human tissues in Human Protein Atlas (HPA) was also updated using the SARS-CoV-2 phosphoproteins as background (Bouhaddou, 2020). Here we selected gene sets with 50-150 SARS-CoV-2 associated phosphoproteins. Different gene-set filters were used for the HPA analysis because the gene expression signatures generally comprised larger gene sets and were not structured hierarchically like GO. We also limited our analysis to “High” gene expression signatures in HPA and excluded “Medium” and “Low” signatures.

Fortunately, we recovered most of the tissues we reported previously (30 tissues were found here; 25 of those were reported in the previous revision; see new **Figure 2E** below). Note that the X-axis of the left panel is different: now we list all genes with pSNVs in various tissues while previously we listed the genes at the individual peak enrichment points of various tissues. The tissues shown on the X-axis are now ranked from the strongest to the weakest enrichment of expression signature.

Lastly, we apologise for the miscommunication. We agree with the Reviewer that limited statistical power should not be the sole rationale to discard a statistical test. The tests should be appropriate to the scientific question that is being asked.

RE: Manuscript MSB-2021-10823R, Human phospho-signaling networks of SARS-CoV-2 infection are rewired by population genetic variants

To clarify, our analysis works at the intersection of phosphoproteomics and human disease genetics. Therefore, the background set of genes/proteins used in our analyses depends on what scientific question is being asked. We would politely suggest that both approaches are valid under some circumstances, depending on the hypothesis at hand:

- For a phosphoproteomics study, the enrichment analysis relative to SARS-CoV-2 phosphoproteins is justified, as recommended above. This statistical test would answer the question:
 - “What pathways and processes are the potential disease-associated genes generally involved in, given that all genes are involved in SARS-CoV-2 phosphorylation signalling?”
- Our study also includes exome-wide genetic variation data and is relevant to the human genetics research community. In a human genetics study, using all phosphoproteins in the human proteome as background would also be appropriate in our opinion, as we did in the initial analysis. This other statistical test would answer a different question:
 - “What pathways and processes are these candidate disease-associated genes generally involved in, given that all these genes are involved in phosphorylation signalling?”

Fortunately, as we show above, both approaches deliver comparable results, indicating that the identified biological signal is robust. We are happy to provide our results with a more-stringent background gene universe, as requested in this comment.

We have updated our Methods section and added more details on this analysis.

2. The statistical testing done in this work is at times either not supporting the conclusions or not described sufficiently. It is unclear to me how the permutations for motif switching were done exactly. The methods are not precise enough here. (The following statement was added: "The expected numbers of other functional annotations we assessed similarly: all motif switching pSNVs, CDK/MAPK to TBK1 motif switches, and pSNVs in kinase binding sites.") Is a given CDK/MAPK motif more likely to switch to TBK if the protein is responsive to a SARS-COV-2 infection or are just more proteins with CDK/MAPK motifs responsive to a SARS-COV-2 infection on the phospho level? A simple Fisher's test might be sufficient instead of permutations.

In my previous review I was quite excited about the observation that MAPK/ERK1 motifs were consistently switching to IKK and PIKK motifs. However, one has to carefully assess if this could happen by chance. Obviously, all of the motifs are very similar, since switching a single site suffices to switch the motif. If in the relevant protein set there are many more MAPK/ERK1 motifs compared to IKK motifs to begin with, it is more likely (simply by chance) that they get 'hit' by a polymorphism, which might switch that motif towards IKK. The description in the Methods section is very slim and does not clarify if this potential bias has been considered and corrected for.

RE: Manuscript MSB-2021-10823R, Human phospho-signaling networks of SARS-CoV-2 infection are rewired by population genetic variants

Thanks for the comment. It appears that the primary concern is the lack of details in the Methods section. We apologise for this limitation and have significantly extended the description of this procedure in Methods (please see the section copied further below).

Based on Comment #1, we believe that the reviewer agrees that our procedure is statistically sound for evaluating the properties of pSNVs (segment pasted below).

"I do agree with the authors that in cases where SNVs inside proteins with differentially phosphorylated sites are compared to the background of SNVs in phosphoproteins such testing is okay".

Briefly, we developed an end-to-end procedure for the permutation test that precisely replicates the selection of control pSNVs (cpSNVs) and their functional annotation with respect to motifs, kinase binding sites and other properties, with exactly the same procedures that we used for the SARS-CoV-2 associated phosphosites. The only "moving parts" in this procedure are a) the use of SARS-CoV-2 associated phosphosites for the main signals, and b) the use of sampled phosphosites from the human proteome as controls, executed for 10,000 sampled iterations. The control phosphosites are sampled in the same numbers (n=1530) as the SARS-CoV-2 associated sites detected in the study (Bouhaddou,2020).

Once we execute the procedure for the SARS-CoV-2 associated phosphosites and the 10,000 sets of random control phosphosites, we examine the control phosphosites to evaluate multiple characteristics of SARS-CoV-2 associated phosphosites. Note that the control phosphosites remain the same for the various analyses.

- Direct impact on phosphoresidues, motif-rewiring, distal and proximal impact (**Figure 1B**),
- Overlap of pSNVs and known binding sites of kinases (**Figure 1D**),
- Motif-rewiring impact focusing on MAPK, CDK and TBK1 motifs (**Figure 4D**).

We believe that this procedure is as appropriate (or more appropriate) for our scenario compared to a Fisher's exact test that is suggested above. One advantage of the end-to-end permutation approach is that it accurately captures the complex workflow we use to select and annotate pSNVs. As such, our approach relies less on the statistical assumptions of a particular statistical test. The permutation test is also more conservative in statistical significance estimates and less inflated in better-powered estimates (e.g., larger categories of pSNVs), making it more appropriate for our use since we only perform a limited number of permutations.

The second part of the comment suggests some additional potential biases. We politely suggest that these biases are unfounded in the context of our research question, as the reviewer pointed out previously. Our process is valid because we compare SNVs in proteins with differentially phosphorylated sites to the background of all SNVs in phosphoproteins.

RE: Manuscript MSB-2021-10823R, Human phospho-signaling networks of SARS-CoV-2 infection are rewired by population genetic variants

We are predominantly interested in deciphering human genetic variation that may contribute to disease phenotypes and risk of severe disease by modifying underlying signalling networks. Therefore, the appropriate control set for our research question is the background set of phosphosites randomly sampled from the phosphoproteome.

For example, even if certain short and degenerate kinase sequence motifs are more frequent among SARS-CoV-2 associated phosphosites compared to the phosphosites in the overall phosphoproteome, the pSNVs enriched in these disease-related phosphosites highlight an important feature of SARS-CoV-2 infection pathways and their interactions with human genetic variability. Further studies of these pSNVs and their role in phospho-signalling may improve our understanding of disease mechanisms and evolution of antiviral defense responses in humans. We have added one sentence in the discussion section to mention the caveat.

Our permutation analysis of pSNV impact considers sites from the entire human phosphoproteome as controls, thus it also emphasizes features of pSNVs that are characteristic to the SARS-CoV-2-responsive signalling network.

Lastly, please see the improved Methods copied below.

Enrichment of motif-rewiring pSNVs. Custom permutation tests were used to evaluate the expected numbers of pSNVs with functional impact predictions (direct, motif-rewiring, proximal, distal) and other pSNV annotations. As controls, we randomly sampled 10,000 sets of phosphosites from the human phosphoproteome such that each sample contained the same number of phosphosites (1530) as was detected in the SARS-CoV-2 infection experiment. The control phosphosites in the human proteome were retrieved from ActiveDriverDB and excluded the SARS-CoV-2 associated phosphosites. The control sites were annotated using gnomAD SNVs and MIMP analysis identically to the SARS-CoV-2 associated phosphosites analysed elsewhere in the manuscript, and control pSNVs (cpSNVs) were derived. The impact of cpSNVs in control phosphosites, their motif-rewiring annotations, and overlap with known kinase binding sites were also derived identically. To derive the statistical significance of pSNV impact in SARS-CoV-2-associated human phosphosites (i.e., direct, distal, proximal, motif-rewiring), we counted how many times the cpSNVs in the control phosphosites were observed with these impacts across the 10,000 iterations. Two empirical P-values were computed as the fractions of times the randomly sampled phosphosites produced greater or smaller counts of cpSNVs with similar impact (i.e., $P = N / 10,000$), and the smaller of the two P-values was reported as the final P-value reflecting a one-tailed test to measure either the over- or under-representation of impact annotations. The expected counts of cpSNVs with specific impacts were derived from the randomly sampled phosphosites and were shown as mean values with ± 1 standard deviation (s.d.) for confidence intervals. We also used precisely the same strategy and the same randomly sampled sets of control phosphosites to evaluate the significance of additional pSNV annotations over-represented in SARS-CoV-2 associated phosphosites. First, a subset of SARS-CoV-2 associated phosphosites occurred in previously annotated binding sites of certain kinases. For each such kinase, we asked how often the cpSNVs in the 10,000 sets of control phosphosites occurred in the known binding sites of the kinase as sampled from the human phosphoproteome. Kinases with over-represented pSNVs in the SARS-CoV-2 associated phosphosites were shown if their frequencies significantly exceeded the expected frequencies of cpSNVs in known kinases binding sites among the 10,000 sets of control phosphosites (empirical $P < 0.05$). Second, we counted the pSNVs in SARS-CoV-2 associated phosphosites for which motif-switching impact was predicted (i.e., loss of a sequence motif combined with gain of another motif). To evaluate the enrichment of these pSNVs relative to the human phosphoproteome, we counted the equivalent motif-rewiring cpSNVs in each of the 10,000 sets of control phosphosites sampled from the human proteome and reported the corresponding empirical P-values as described above. Third, we focused on the subset of motif-switching pSNVs in SARS-CoV-2 associated phosphosites that led to losses of CDK/MAPK sequence motifs and simultaneously induced TBK1 motifs. Equivalent counts of CDK/MAPK-to-TBK1 motif-switching cpSNVs were derived from the same 10,000 sets of control phosphosites sampled from the human proteome. We reported the corresponding empirical P-values as described above.

RE: Manuscript MSB-2021-10823R, Human phospho-signaling networks of SARS-CoV-2 infection are rewired by population genetic variants

3. The authors tend to use the word "affect" loosely. It should only be used to report known facts. A phosphosite being predicted to switch a motif, doesn't per se justify the usage of "affect" (e.g. line 286) when indicating that a pSNV is located at a phosphosite. In my interpretation the use of the word "affect" implies that the SNV was shown to change the phosphorylation of this site. In this work effects on phosphorylation are predicted and not validated, therefore I would advise against the use of this phrase in this context. In the case of phosphosites where the phosphorylated residue itself is mutated it would be acceptable to use the term.

Thanks for this constructive feedback. We agree that more cautious language is justified, especially in this rapidly evolving field of research and in an area of public health. Most of our team are not native English speakers and admittedly these mistakes happen sometimes, especially when writing collaboratively and on tight timelines.

We replaced most of the instances of 'affect' in the manuscript.

Minor points

4. Regarding point 1 from Reviewer 1: Here the reviewer was asking to compare against phospho data from other viruses. The authors' reply was as follows: "Thank for a great point. We now compared SARS-CoV-2 associated phosphosites with two additional sets of phosphosites detected in human immunodeficiency virus (HIV) infection and in herpesvirus (HSV-1) infection. In both cases, the other phosphosites were mostly distinct from SARS-CoV-2 associated phosphosites. Approximately 17% of SARS-CoV-2 associated sites were shared, and ~15% of sites with pSNVs were shared. Therefore, our findings are relatively specific to SARS-CoV-2 infections. This analysis has been added to Figure 1F-G."

I appreciate the inclusion of additional data in the analysis. However, the first observation (only 17% of the phospho-sites overlapped with the other data) is a feature of the originally published phospho-proteomics data, i.e. unrelated to the question of genomic variability. The second result (15% with pSNVs were shared; i.e. 88% out of the 17% from above) is not surprising in light of the fact that 73% of the phospho-sites have a pSNV. 88% is still more than 73%, but it is unclear if this apparent enrichment could happen by chance.

We conducted this analysis and found that SARS-CoV-2 phosphosites with pSNVs are depleted among the phosphosites shared with other virus infections, contrary to the expectation of the Reviewer who suggested an enrichment. Details are shown below.

In the second paragraph, one of the numbers provided by the Reviewer appears to be incorrect. The fraction of SARS-CoV-2 sites with any pSNVs is 65% (not 73% as indicated in the comment). The correct fraction of phosphosites with pSNVs is derived from the numbers 987 (shown in line 119) and 1530 (shown in line 110).

RE: Manuscript MSB-2021-10823R, Human phospho-signaling networks of SARS-CoV-2 infection are rewired by population genetic variants

$$\frac{987}{1530} = 0.645098$$

There are 17% of phosphosites shared with SARS-CoV-2 and the two other viruses, shown in **Figure 1F**. $127 + 108 + 25 = 260$

$$\frac{260}{1530} = 0.1699346$$

There are 15% mutated phosphosites shared with SARS-CoV-2 and the two other viruses, shown in **Figure 1G**. $73 + 64 + 15 = 152$

$$\frac{152}{987} = 0.154002$$

Relative to all SARS-CoV-2 sites, the fraction of shared and mutated sites is smaller.

$$\frac{152}{1530} = 0.09934641$$

Expected fraction of shared and mutated sites would be the following.

$$0.645098 * 0.1699346 = 0.1096245$$

We can derive the expected count of shared and mutated sites and its confidence intervals by sampling. Note that the observed count 152 is lower than the 95% confidence interval.

$$\begin{aligned} \text{Expected count} &= 168 \\ 95\% \text{ confidence interval} &= 153.975 - 182 \end{aligned}$$

We can confirm this finding using Fisher's exact tests, with the following association table. The vertical axis shows SARS-CoV-2-associated phosphosites not shared or shared with other infections and the horizontal axis shows SARS-CoV-2-associated phosphosites without or with pSNVs.

#		FALSE	TRUE
# FALSE		435	835
# TRUE		108	152

Two-tailed Fisher's exact test:

$$\text{p-value} = 0.02746155$$

One-tailed Fisher's exact test, alternative hypothesis of greater than expected:

$$\text{p-value} = 0.9890306$$

One-tailed Fisher's exact test, alternative hypothesis of lower than expected:

RE: Manuscript MSB-2021-10823R, Human phospho-signaling networks of SARS-CoV-2 infection are rewired by population genetic variants

p-value = 0.01571787

Therefore, the overlap of

a) mutated SARS-CoV-2 phosphosites, and
b) SARS-CoV-2 phosphosites shared with other virus infections,
are lower than expected. This suggests that the pSNVs are mostly specific to SARS-CoV-2 infection, and by extension, so are most of the findings described in this manuscript.

We added this result to our manuscript and hopefully improved the wording to better convey the way the significance was calculated.

There were 152 SARS-CoV-2 associated sites with pSNVs that were shared with other infections (10% of 1530), fewer than expected from chance (168 expected, Fisher's exact $P = 0.027$) (Figure 1F-G).

5. In Figure 1a the diamonds symbols are only present in the legend.

There is no special meaning to the diamonds, they only indicate the colors of pSNV impact. The colors reflect the colors of **Figure 1B**. We replaced diamonds with rectangles to improve the clarity of the figure.

6. In Figure 1d all shown kinases are overrepresented. How can that be? Does the set of responsive phosphoproteins have more high quality motifs in general? If yes that has to be controlled.

Only the top-15 kinases with the most frequent pSNVs were shown in **Figure 1D**, as was specified in the figure caption.

We updated this figure to show kinases with $P < 0.05$ (permutation test). This recovers 12 kinases, of which 11 were shown previously and one kinase was added (PAK2). This indicates that the kinases with the largest number of pSNVs in well-studied binding sites are also statistically over-represented in pSNVs. Please see new **Figure 1D** below.

We did not show the full list of kinases because the figure would have been too large. We would be happy to provide it as a supplemental figure, however it seems like a small point since our primary goal is to show statistically significant results.

Regarding the second part of the comment on motifs in SARS-CoV-2 responsive phosphoproteins: this analysis is not about motifs. Instead, this is about well-defined phosphosites for which there is site-specific evidence of specific kinases binding the sites, as documented in proteomics databases.

The statistical procedure in this analysis is the same that is discussed in Comment #2 that was assessed favourably in Comment #1 above.

The comment appears to suggest an alternative explanation that SARS-CoV-2-responsive phosphosites may be better described in the databases than phosphosites in general. While this may be the case, we are confident that the background set of all proteome-wide phosphosites is an appropriate control for our research question.

RE: Manuscript MSB-2021-10823R, Human phospho-signaling networks of SARS-CoV-2 infection are rewired by population genetic variants

The control of proteome-wide phosphosites is appropriate here because it directly addresses disease-related genetic variation in an optimal way. We find that genetic polymorphisms (pSNVs) in this SARS-CoV-2-associated part of the signalling network occur in the well-defined target sites of those kinases more often than would be expected in similar sets of phosphosites proteome-wide, increasing our confidence in these phosphosites and the respective pSNVs.

7. A sentence in line 231 seems incoherent: "Genes with frequent pSNVs genes"

8. A word is missing in line 281: "TBC1D4 (i.e., AS160) is a GTPase-activating protein"

Thanks for this detailed review! We fixed the typos.

9. I fail to grasp the argument in line 280 and following. Are the authors arguing that the pSNV increases the risk of severe COVID through an increase in the risk of diabetes type II?

Thank you for pointing out the lack of clarity. We prefer to refrain from a direct speculation of causality of the roles of pSNVs in COVID-19, diabetes and/or obesity.

This section is a synthesis of different observations:

- a) some phosphosites of SARS-CoV-2 infection and a known disease pathway are shared,
- b) the diseases of the pathway are comorbidities of COVID-19,
- c) some pSNVs have strong evidence of altering phosphosites and pathway activity,
- d) previous genetic data implicates other SNVs in these genes in the obesity and diabetes, and
- e) the pSNVs are infrequent in the human population in the gnomAD database (not all gnomAD individuals are confirmed to be healthy and some may have diabetes, but that level of data is not available to us).

It is interesting that the signaling networks of diabetes and SARS-CoV-2 infection are shared, and there are pSNVs involved in this intersection with clear mechanistic and epidemiological hypotheses to follow up on in future studies.

We improved the language in this section cautiously, by adding a sentence at the end of the section to convey this message.

We found that the phosphorylation networks of SARS-CoV-2 infection also affect phosphosites in a molecular pathway involved in human physiology and diseases that have been implicated as comorbidities of COVID-19. Furthermore, certain pSNVs in these phosphosites have functional predictions of altering pathway activity, while other SNVs in the genes have been associated with the same disease phenotypes in genetic studies. However, our observations should be interpreted with caution since the data are not sufficient to infer a causal relationship of pSNVs, diabetes or obesity, and COVID-19. In summary, some

RE: Manuscript MSB-2021-10823R, Human phospho-signaling networks of SARS-CoV-2 infection are rewired by population genetic variants

pSNVs provide mechanistic and epidemiological hypotheses for studying the interactions of COVID-19 and human phenotypes and diseases.

10. In line 304 the authors state that 217 pSNVs correspond to 3360 predictions of motif rewiring. This would correspond to around 15 motif rewirings per site. This could be written more clearly.

Thanks, good point. The large number of potential rewiring events per pSNV is caused by the redundancy of underlying predictions in the MIMP software that is due to the similarity of sequence motifs of related kinases. Elsewhere in the manuscript we have only selected the top motif gain and/or top motif loss per pSNV.

We have added the following sentence in the Section to clarify.

Since the individual pSNVs were often associated with several similar sequence motifs in MIMP predictions, this analysis combined 3360 high-confidence motif-rewiring predictions across the pSNVs and resulted in partially redundant functional impact scores.

11. In line 359 the authors state that the gene ARFGEF2 causes several disease phenotypes. Do they mean that a gain, full loss or partial loss of function of ARFGEF2 causes this phenotype? How does the SNV relate to this?

We cited a previous paper (<https://www.nature.com/articles/ng1276>) that describes several SNVs of this gene in diseased families, evaluation of gene expression levels through mouse brain development, and *in vitro* experiments with gene inhibition and mutant gene expression that showed the role of the gene in neural proliferation and migration. This is just one of several studies about neuro-developmental disorders in this gene and we politely suggest that a detailed review of this locus is beyond the scope of our study manuscript.

The pSNV in ARFGEF2 that we find is different from the SNVs that were found in the study above. However, the risk-associated pSNV R232H that we find is shown as Variant of Unknown Significance (VUS) in the ClinVar database (Table EV9).

We added a few lines to this section as shown below:

Other ARFGEF2 mutations have been linked to autosomal recessive periventricular heterotopia with microcephaly (ARPHM), a rare disorder involving cerebral malformations, severe developmental delay, and recurrent pulmonary infections, whereas loss-of-function experiments implicated the gene in neuronal proliferation and migration⁶⁴. The R232H pSNV is indicated as a variant of unknown significance for ARPHM in the ClinVar database (Table EV9).

12. How can the pSNV P257L alter a phosphosite (a serine) also at position S257 (lines 391-392)

The phosphosite should be S256 – thanks for a great catch!

Thank you for your message asking us to reconsider our decision on your manuscript MSB-2022-10825, and thank you for sending us your revised manuscript and point-by-point response. We have now heard back from Reviewer #2, who agreed to evaluate the revisions and responses. As you will see below, this reviewer is satisfied with the modifications made and thinks that the study is now suitable for publication.

Before we can formally accept your manuscript, we would ask you to address the following issues.

Comments from Reviewer #2 :

"The authors have changed the statistical analysis of gene enrichment and now it much better fits the purpose. In their response letter the authors argue that both - their old and the new - enrichment analyses would be appropriate. I still have issues with that reasoning, because I think they actually test different things, i.e. they are not equivalent. However, since the authors have changed that part of the analysis, the manuscript is fine now.

Minor:

Figure 4: Panel E is shown between B and C. Further, the description (caption) of panel E does not seem to fit the panel."

Comments from Reviewer #2 :

"The authors have changed the statistical analysis of gene enrichment and now it much better fits the purpose. In their response letter the authors argue that both - their old and the new - enrichment analyses would be appropriate. I still have issues with that reasoning, because I think they actually test different things, i.e. they are not equivalent. However, since the authors have changed that part of the analysis, the manuscript is fine now.

Thank you for the positive comments and approving our manuscript for publication.

Minor:

Figure 4: Panel E is shown between B and C. Further, the description (caption) of panel E does not seem to fit the panel."

Thanks for the comment and great catch. We rearranged the panels to correspond to the flow in the manuscript (panels D and E were switched in the figure). The caption was also updated accordingly.

Thank you again for sending us your revised manuscript. We are now satisfied with the modifications made and I am pleased to inform you that your paper has been accepted for publication.